# Hypothalamic CDK4 regulates thermogenesis by modulating sympathetic innervation of adipose tissues

Judit Castillo-Armengol[1] (ID), Valentin Barquissau[1], Sarah Geller[1], Honglei Ji[1], Ilenia Severi[2], Wiebe Venema[2], Eric Aria Fenandez[1], Catherine Moret[1], Katharina Huber[1], Lucia C Leal-Esteban[1], Anita Nasrallah[1], Laia Martinez-Carreres[1], Guy Niederhäuser[1], Patricia Seoane-Collazo[3,4], Sylviane Lagarrigue[5], Miguel López[3,4] (ID), Antonio Giordano[2], Sophie Croizier[1] (ID), Bernard Thorens[1], Isabel C Lopez-Mejia[1] & Lluis Fajas[1,*] (ID)

## Abstract

This study investigated the role of CDK4 in the oxidative metabolism of brown adipose tissue (BAT). BAT from $Cdk4^{-/-}$ mice exhibited fewer lipids and increased mitochondrial volume and expression of canonical thermogenic genes, rendering these mice more resistant to cold exposure. Interestingly, these effects were not BAT cell-autonomous but rather driven by increased sympathetic innervation. In particular, the ventromedial hypothalamus (VMH) is known to modulate BAT activation via the sympathetic nervous system. We thus examined the effects of VMH neuron-specific $Cdk4$ deletion. These mice display increased sympathetic innervation and enhanced cold tolerance, similar to $Cdk4^{-/-}$ mice, in addition to browning of scWAT. Overall, we provide evidence showing that CDK4 modulates thermogenesis by regulating sympathetic innervation of adipose tissue depots through hypothalamic nuclei, including the VMH. This demonstrates that CDK4 not only negatively regulates oxidative pathways, but also modulates the central regulation of metabolism through its action in the brain.

**Keywords** adrenergic innervation; brown adipose tissue; CDK4; hypothalamus; mitochondria; thermogenesis
**Subject Categories** Metabolism; Neuroscience

## Introduction

Growing evidence demonstrates that a regulatory crosstalk exists between metabolic pathways and regulators of cell cycle progression. Our laboratory and others have demonstrated that CDK4 is one of such "metabolic" cell cycle regulators (Blanchet *et al*, 2011; Denechaud *et al*, 2016; Lagarrigue *et al*, 2016; Lopez-Mejia *et al*, 2017). CDK4 participates in the G1/S transition phase of the cell cycle. Growth factors or mitotic stimuli activate this kinase by promoting its interaction with D-type cyclins, ultimately resulting in the activation of the E2F transcription factors, which regulate the expression of genes important for DNA synthesis and chromosome duplication (Malumbres, 2014). In addition to, or in complement to, cell cycle regulation, CDK4 represses the oxidative metabolism indirectly, via the control of the E2F1 transcription factor in muscle and brown adipose tissue (BAT) (Blanchet *et al*, 2011), and promotes the insulin signaling pathway in mature adipocytes (Lagarrigue *et al*, 2016). Interestingly, we previously showed that the AMP-activated protein kinase (AMPK), which is a central inhibitor of such anabolic processes, is a direct target of, and is repressed by, CDK4 (Lopez-Mejia *et al*, 2017). Overall, the participation of cell cycle regulators in the control of energy homeostasis occurs mainly through the activation of anabolic processes and the repression of the oxidative metabolism.

Because of the role of CDK4 in the regulation of mitochondrial activity and in the overall oxidative metabolism in MEFs (Lopez-Mejia *et al*, 2017), we investigated the function of CDK4 in BAT biology, which is a tissue in which oxidative metabolism is central for its thermogenic function.

Adipose tissue has a major contribution to the regulation of energy homeostasis. Distinct adipose tissue types have different

1   Center for Integrative Genomics, University of Lausanne, Lausanne, Switzerland
2   Department of Experimental and Clinical Medicine, Marche Polytechnic University, Ancona, Italy
3   NeuroObesity Group, Department of Physiology, CIMUS, University of Santiago de Compostela-Instituto de Investigación Sanitaria, Santiago de Compostela, Spain
4   CIBER Fisiopatología de la Obesidad y Nutrición (CIBEROBN), Santiago de Compostela, Spain
5   Department of Physiology, University of Lausanne, Lausanne, Switzerland
    *Corresponding author. Tel: +41 21 692 41 11; E-mail: lluis.fajas@unil.ch

functions in the overall regulation of metabolism. Essentially, three classes of adipose tissue can be distinguished: white adipose tissue (WAT), which stores energy in the form of triacylglycerols (TAG), BAT that dissipates energy in the form of heat, and beige adipose tissue that also generates heat, but in contrast to BAT is embedded into WAT, and has a specific gene expression pattern (Giralt & Villarroya, 2013).

Brown adipose tissue is unique in that its activation can increase energy expenditure and glucose clearance (Lowell *et al*, 1993; Harper *et al*, 2001; Cannon & Nedergaard, 2009). BAT is a highly specialized thermogenic organ which acts to maintain body temperature during cold stress (Cannon & Nedergaard, 2004) and is composed of brown adipocytes that contain numerous lipid droplets and a high number of mitochondria which express uncoupling protein 1 (UCP1). UCP1 uncouples ATP production from mitochondrial respiration to produce heat (Cannon & Nedergaard, 2004; Fedorenko *et al*, 2012). These characteristics suggest that the modulation of BAT activity may be harnessed therapeutically to combat obesity and diabetes. Although initially described in small mammals in the 16th century, evidence of metabolically active BAT depots in adult humans only came to light more recently (Nedergaard *et al*, 2007).

Cold perception is mediated by sensory neurons found in several parts of the body, such as the skin, which in turn relays their information to the central nervous system (CNS) (Morrison *et al*, 2012, 2014). There, several hypothalamic nuclei integrate this sensory information to regulate the thermogenic response (Oldfield *et al*, 2002; Monda *et al*, 2005; Morrison *et al*, 2012; Contreras *et al*, 2015; Martins *et al*, 2016; Lee *et al*, 2018). Among them, the ventromedial nucleus of the hypothalamus (VMH) was one of the first nuclei described as a regulator of thermogenesis (Perkins *et al*, 1981). Indeed, a major population of VMH neurons which express the transcription factor steroidogenic factor-1 (SF1) has been shown to contribute to BAT activation, and thus thermogenesis, by modulating its sympathetic stimulation (Kim *et al*, 2011; Contreras *et al*, 2015; Labbe *et al*, 2015; Martinez-Sanchez *et al*, 2017; Seoane-Collazo *et al*, 2018). The sympathetic nervous system directly innervates BAT to control heat production (De Matteis *et al*, 1998; Murano *et al*, 2009; Bartness *et al*, 2010; Lopez *et al*, 2010). Tyrosine hydroxylase-expressing nervous fibers (TH fibers) in BAT secrete norepinephrine, the principal stimulator of BAT activity, locally (Bartness *et al*, 2010). Norepinephrine in turn stimulates β-adrenergic receptors in brown adipocytes to increase intracellular cAMP levels via the adenylate cyclase and activate the PKA pathway, thus increasing lipid mobilization and thermogenesis (Collins

& Surwit, 2001). Moreover, sympathetic innervation also increases mitochondrial number and activity (Klingenspor *et al*, 1996; Puigserver *et al*, 1998; Sanchez-Alavez *et al*, 2013).

In the present study, we show that *Cdk4* knockout mice (*Cdk4*nc/nc, referred to in this manuscript as *Cdk4*−/−) (Martin *et al*, 2003; Lagarrigue *et al*, 2016) are resistant to cold. By using two additional mouse models, in which we deleted *Cdk4* specifically in BAT or in the VMH SF1 neurons, we prove that these effects are mediated by the action of CDK4 in the hypothalamus, and raise the possibility of a broader function for CDK4 in other brain areas.

# Results

## *Cdk4* knockout mice show increased energy expenditure, improved adaptation to cold, and increased mitochondrial volume

*Cdk4* knockout mice (*Cdk4*−/−) were first used to explore the role of CDK4 in BAT metabolism. As previously described, *Cdk4*−/− mice were smaller (Fig EV1A) and had decreased body mass and decreased fat mass compared to *Cdk4*+/+ mice (Fig EV1B and C) (Lagarrigue *et al*, 2016). Despite their smaller size and leaner phenotype, these mice showed a 20% increase in food intake compared to *Cdk4*+/+ mice when corrected by body weight (Fig EV1D and E). Moreover, relative to control mice, at room temperature (24°C), *Cdk4*−/− mice showed a significant increase in oxygen consumption (Fig 1A and B) and energy expenditure (EE; Fig 1C and D). The respiratory exchange ratio (RER) was slightly decreased, namely during the dark phase (Fig 1E and F), in *Cdk4*−/− compared to *Cdk4*+/+ mice, suggesting that these mice preferentially used fatty acids as energy substrates.

Considering that the activation of BAT is known to increase whole-body energy expenditure by increasing thermogenesis (Cannon & Nedergaard, 2009), we hypothesized that the increased EE we observed in the *Cdk4*−/− mice could be explained by increased thermogenesis. Indeed, BAT activity is specifically increased during cold exposure in order to maintain body temperature. Thus, to explore the activation of BAT in *Cdk4*−/− mice, we exposed the mice to a 6°C challenge and measured oxygen consumption, EE, and the RER by indirect calorimetry. Cold exposure (from 24 to 6°C) increased oxygen consumption (Fig 1A and B) and EE (Fig 1C and D), and decreased RER (Fig 1E and F) in the control *Cdk4*+/+ animals. Compared to controls, at 6°C *Cdk4*−/− mice showed higher levels of oxygen consumption (Fig 1A and B), increased EE (Fig 1C

**Figure 1. Increased whole-body oxygen consumption and lipid utilization, and improved resistance to cold exposure in *Cdk4*−/− mice.**

A–F   Indirect calorimetry was performed using the Oxymax/CLAMS system in *Cdk4*+/+ (n = 6) and *Cdk4*−/− (n = 5) mice. Whole-body oxygen consumption rate (VO2) (A, B), energy expenditure (EE) (C, D), and respiratory exchange ratio (RER) (E, F) were measured at 24°C and during a cold challenge at 6°C during both the light (white rectangle) and dark (black rectangle) phases over a 2-day period.

G–I   CalR was used to implement GLM-regression plot for each group corresponding to the association between energy balance and total body mass at 24°C (G) and the association between energy balance and total body mass at 6°C (H). The CalR interface displays each mouse as a dot, and the standard error of mean for each group in gray. The "mass effect" and "group effect" were analyzed using a generalized linear model (GLM) using body weight as a covariate. The results of this analysis are shown in table (I) for *Cdk4*+/+ (n = 6) and *Cdk4*−/− (n = 5) mice.

J, K   Basal rectal temperature was measured at 24°C (J), and rectal temperature was monitored during acute cold exposure at 4°C for 6 h (K) in *Cdk4*+/+ (n = 10) and *Cdk4*−/− (n = 8) mice.

Data information: All data are shown as the mean ± SEM; Student's *t*-test was used for statistical analysis. *P < 0.05, **P < 0.01, ***P < 0.001.
Source data are available online for this figure.

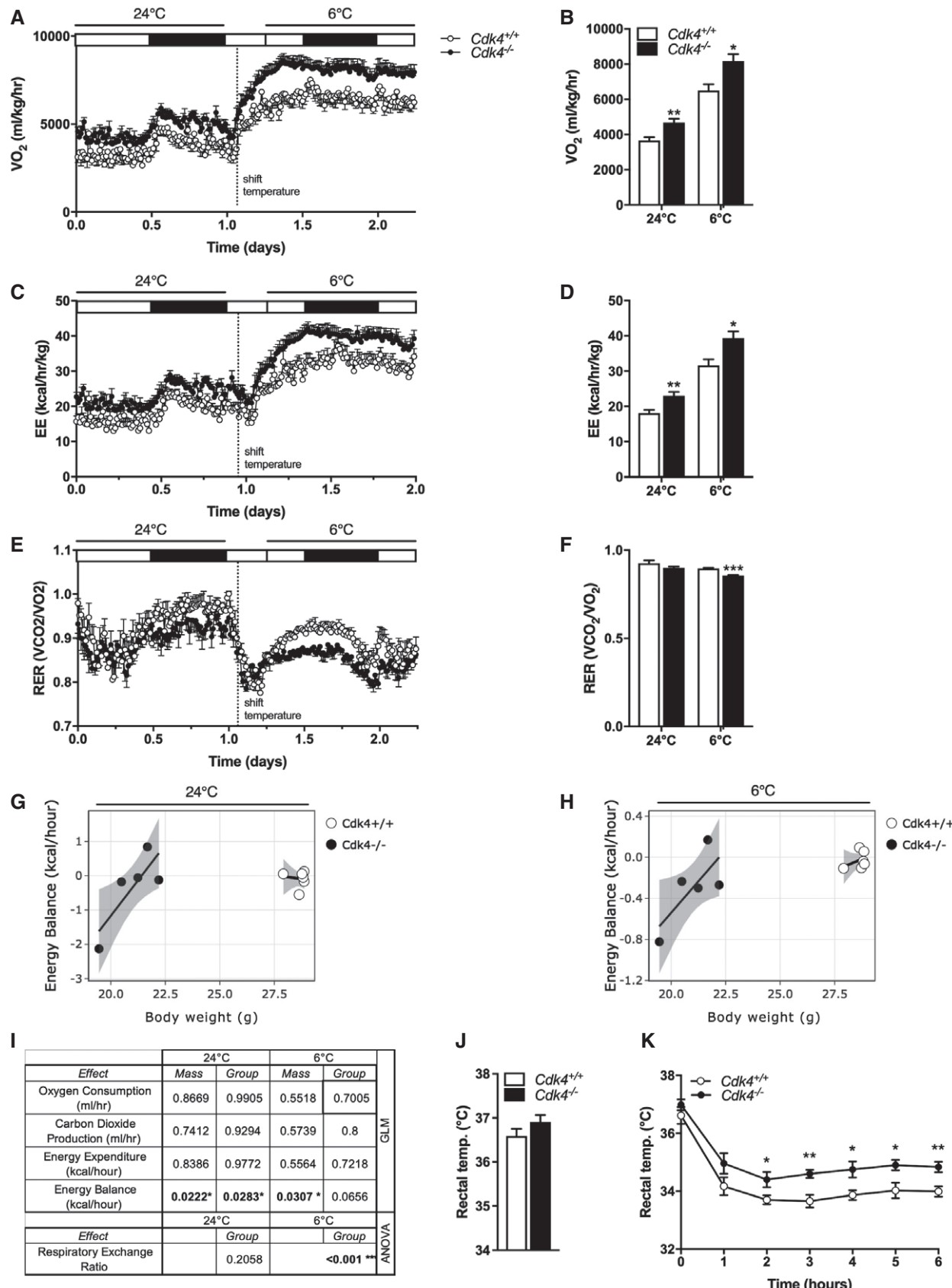

**Figure 1.**

and D), and a greater decrease in RER (Fig 1E and F), indicating increased lipid utilization. In addition, when the energy balance was calculated, $Cdk4^{-/-}$ mice clearly displayed a negative energy balance, at room temperature and upon cold exposure (Fig 1G and H), which is significantly different from the behavior observed in $Cdk4^{+/+}$ mice (Fig 1I). These data suggested that $Cdk4^{-/-}$ mice are primed to better respond to energy stress situations, like cold exposure than their control littermates. To test this hypothesis, we next exposed the animals to acute cold (4°C) and followed their body temperature for 6 h. The basal rectal temperature of $Cdk4^{-/-}$ mice was similar to that of their control littermates (Fig 1J). However, $Cdk4^{-/-}$ mice maintained a consistently higher body temperature than controls during the 6 h of cold exposure (Fig 1K). These results suggest that the depletion of CDK4 results in increased thermogenesis in response to cold.

To understand how CDK4 depletion affects thermogenesis, we next examined the effect of global $Cdk4$ deletion on the morphology and thermogenic activity of interscapular BAT (iBAT). In $Cdk4^{-/-}$ mice, the iBAT mass was decreased and the tissue was visibly darker than in controls (Fig 2A and B), characteristics suggestive of higher oxidative activity. Indeed, morphological analysis of hematoxylin–eosin-stained iBAT revealed a decrease in lipid droplet size in $Cdk4^{-/-}$ animals compared to droplet size in $Cdk4^{+/+}$ mice (Fig 2C). However, analysis of the expression levels of the phosphorylated lipase protein (p-HSL S660) in BAT protein lysates showed no differences, suggesting a compensatory effect in HSL activity due to overactivation (Appendix Fig S3C and D).

We also observed increased expression of UCP1 in the iBAT of $Cdk4^{-/-}$ mice by immunohistochemistry (Fig 2C) and western blot (Fig 2D and E), further suggesting that these animals displayed increased oxidative activity and thermogenesis. Moreover, mRNA level analysis by RT–qPCR revealed a significant upregulation of thermogenic genes and brown adipocyte markers in the $Cdk4^{-/-}$ mice relative to controls, with $Ucp1$, $Dio2$, and $Bmp8b$ being especially highly increased (Fig 2F). Taken together, these results suggested that $Cdk4$-deficient mice display a marked prothermogenic phenotype.

To further investigate potential subcellular metabolic changes induced by $Cdk4$ knockout, we next examined iBAT mitochondrial morphology and function. We observed a significant 40% increase in mitochondrial volume and visibly decreased lipid content in the iBAT of $Cdk4^{-/-}$ mice, as measured by transmission electron microscopy (TEM; Fig 2G and H). Next, we determined the oxidative capacity of iBAT by respirometry analysis using the Oroboros

system. Coupled respiration through complexes I and II (CI + CII) and maximal electron transport system capacity were significantly increased in iBAT homogenates from the $Cdk4^{-/-}$ mice relative to controls (Fig 2I). Taken together, these results suggest that $Cdk4^{-/-}$ mice have a higher thermogenic capacity due to an increased mitochondrial volume and function, which in turn increases the ability of BAT to uncouple mitochondria and generate heat.

## BAT-specific deletion of *Cdk4* has no impact on thermogenesis

To determine whether the deletion of $Cdk4$ from BAT in particular was responsible for the increased thermogenesis observed in $Cdk4^{-/-}$ mice, and to further explore the role of CDK4 in BAT, we generated BAT-specific $Cdk4$ knockout mice by crossing $Cdk4^{flox/flox}$ mice with Ucp1-Cre transgenic mice, which express the Cre recombinase specifically in brown adipocytes, thus generating $Cdk4^{flox/flox}$ Ucp1-Cre mice. $Cdk4^{flox/flox}$ Ucp1-Cre animals, compared to controls, had lower CDK4 levels in iBAT, but normal CDK4 levels in subcutaneous WAT (scWAT) and perigonadal WAT (pgWAT), demonstrating the tissue selectivity of the knockout (Fig EV2A). However, no differences in body size (Fig EV2B), body weight (Fig EV2C), or body composition (Fig EV2D) were observed in $Cdk4^{flox/flox}$ Ucp1-Cre mice compared to those of their control littermates. Moreover, no differences were observed in body temperature at 24°C (Fig 3A) or during cold exposure (4°C for 6 h; Fig 3B) between $Cdk4^{flox/flox}$ Ucp1-Cre and $Cdk4^{flox/flox}$ mice. In contrast to the $Cdk4^{-/-}$ mice, the iBAT mass, and color from $Cdk4^{flox/flox}$ Ucp1-Cre mice, was similar to that of the controls (Fig 3C and D), with no visible differences in the lipid content, UCP1 staining, or the overall morphology of this tissue between the two genotypes (Fig 3E). Likewise, western blot analysis of UCP1 (Fig 3F and G) and RT–qPCR analysis of thermogenic and BAT-specific gene expression levels (Fig 3H) showed no significant difference in protein or gene expression between the two genotypes. These results demonstrated that CDK4 signaling in BAT does not regulate the thermogenic function of brown adipocytes.

## *Cdk4*$^{-/-}$ mice show increased sympathetic innervation of BAT and increased browning of subcutaneous WAT

Since CDK4 signaling does not directly regulate thermogenic function in BAT, we next focused on exploring alternative mechanisms. It has been well established that BAT-mediated thermogenesis is regulated by sympathetic innervation (Bartness *et al*, 2010). TH-positive sympathetic fibers are found in BAT and release

---

**Figure 2. *Cdk4* deficiency promotes thermogenesis and mitochondrial biogenesis in iBAT.**

A    Gross morphology of iBAT from $Cdk4^{+/+}$ and $Cdk4^{-/-}$ mice.

B    iBAT mass normalized to body weight of $Cdk4^{+/+}$ (n = 11) and $Cdk4^{-/-}$ (n = 10) mice.

C    Hematoxylin–eosin staining and UCP1 immunohistochemical (IHC) staining of iBAT sections (scale bar 100 μm) from $Cdk4^{+/+}$ and $Cdk4^{-/-}$ mice.

D, E    Western blot analysis of UCP1 and CDK4 protein expression (D) and quantified protein levels of UCP1 (E) (n = 4 biological replicates) in iBAT. HSP90 was used as loading control.

F    Expression of thermogenic genes and brown adipocyte markers in iBAT of $Cdk4^{+/+}$ (n = 6) and $Cdk4^{-/-}$ (n = 6) mice as assessed by RT–PCR.

G, H    Transmission electron microscopy (TEM) images of iBAT (G) (scale bar 5 μm [upper panel] and 1 μm [lower panel, LD indicates lipid droplet and M mitochondria]) and quantification of lipid droplet number (LD) and mitochondrial volume (Mito) (H) ($Cdk4^{+/+}$ [n = 4] and $Cdk4^{-/-}$ [n = 4]).

I    Respirometry analysis of iBAT as measured by the Oroboros system ($Cdk4^{+/+}$ [n = 6] and $Cdk4^{-/-}$ [n = 6]).

Data information: All data are shown as the mean ± SEM; Student's *t*-test (B, F, H, I) and Mann–Whitney *U*-test (E) were used for statistical analyses.
*P < 0.05, **P < 0.01.
Source data are available online for this figure.

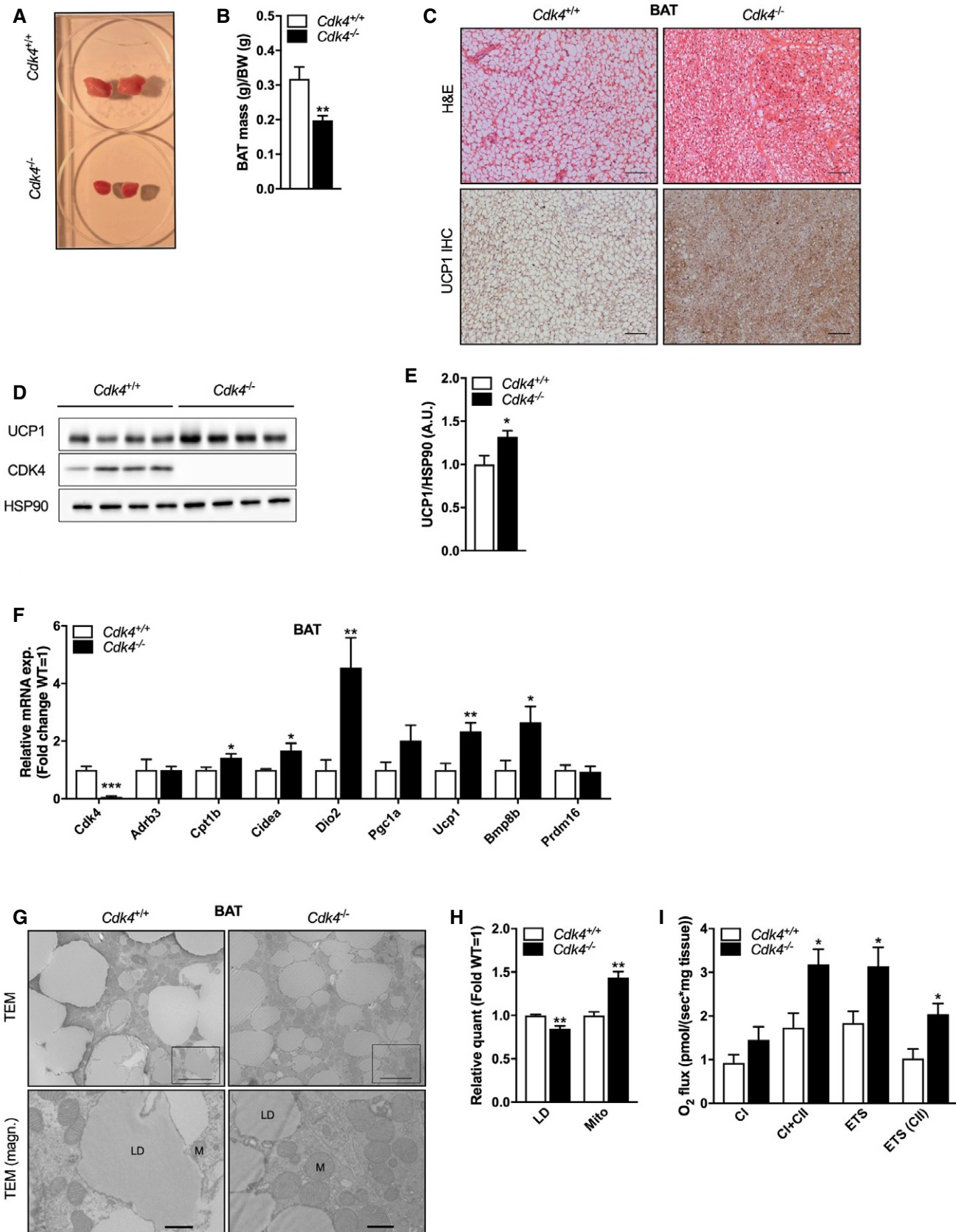

**Figure 2.**

norepinephrine, which is required for BAT activation. We hypothesized that an increase in sympathetic innervation may underlie the increase in EE and thermogenesis observed in the $Cdk4^{-/-}$ mice.

Indeed, $Cdk4$-deficient mice had a significantly higher number of TH-positive fibers in iBAT than wild-type mice, as shown by immunohistochemical analysis (Fig 4A and B). The apparent

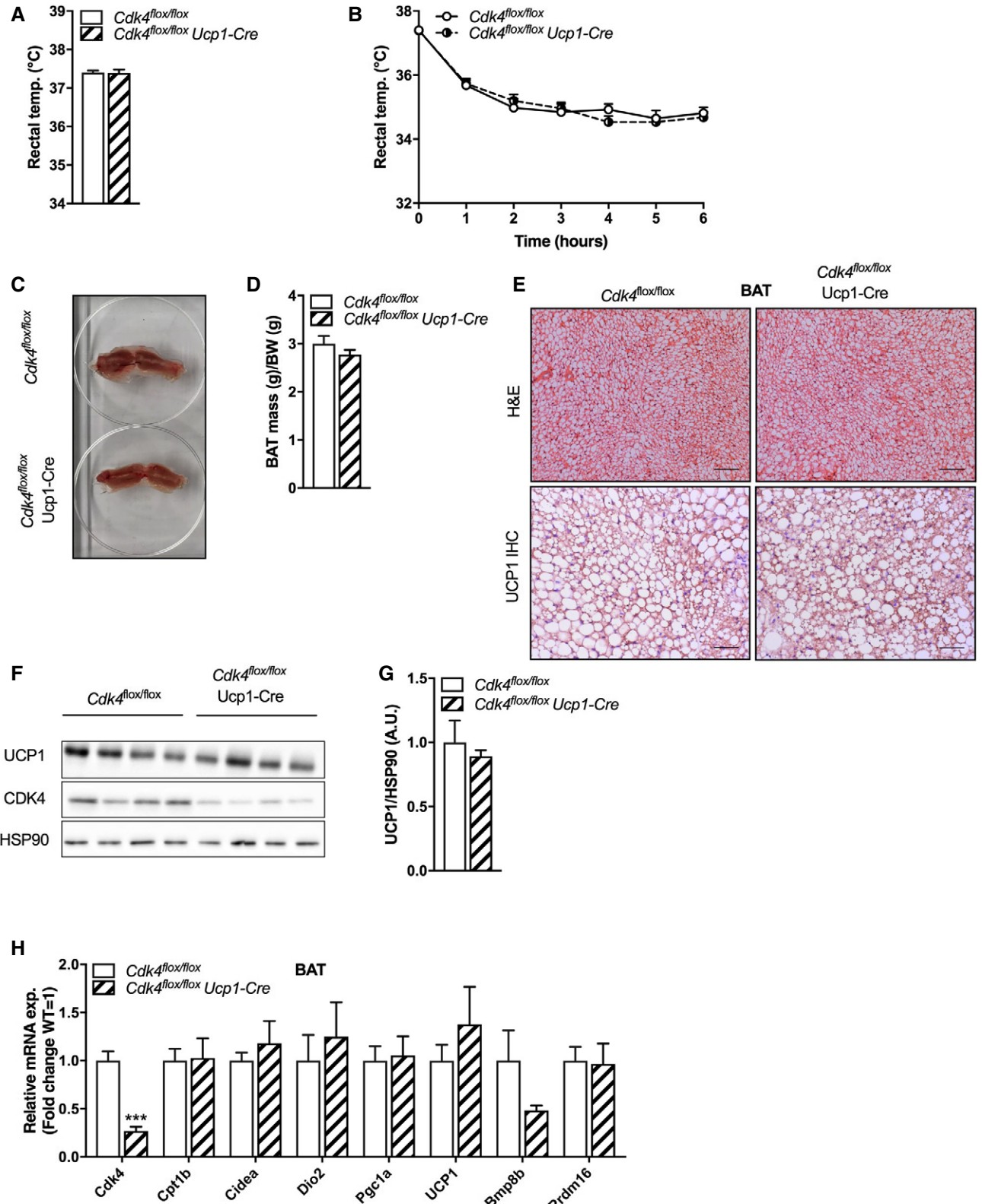

**Figure 3.**

Figure 3. **BAT-specific *Cdk4* deletion does not affect the thermogenic characteristics of iBAT.**

A, B Rectal temperature was measured at 24°C (A), and rectal temperature was monitored for 6 h during acute cold exposure at 4°C (B) in *Cdk4^flox/flox* (*n* = 9) and *Cdk4^flox/flox* Ucp1-Cre mice (*n* = 9).

C Gross morphology of iBAT from *Cdk4^flox/flox* and *Cdk4^flox/flox* Ucp1-Cre mice

D iBAT mass compared to body weight of *Cdk4^flox/flox* (*n* = 8) and *Cdk4^flox/flox* Ucp1-Cre (*n* = 7) mice.

E Hematoxylin–eosin (H&E) staining and UCP1 immunohistochemical (IHC) staining of iBAT sections (scale bar 100 μm [upper panel] and 50 μm [lower panel]) from *Cdk4^flox/flox* and *Cdk4^flox/flox* Ucp1-Cre mice.

F, G Western blot analysis of UCP1 and CDK4 protein expression (F) and quantified protein levels of UCP1 (G) (*n* = 4 biological replicates) in iBAT. HSP90 was used as a loading control.

H Expression of thermogenic genes and brown adipocyte markers in iBAT of *Cdk4^flox/flox* (*n* = 5) and *Cdk4^flox/flox* Ucp1-Cre (*n* = 6) mice as assessed by RT–PCR.

Data information: All data are shown as the mean ± SEM. Student's *t*-test was used for statistical analysis. ***$P < 0.001$.
Source data are available online for this figure.

decrease of TH protein levels found when analyzed its expression by western blotting (Appendix Fig S3A and B) is explained by the unbalance in protein concentration which is notably higher in the BAT of Cdk4$^{-/-}$ mice (Appendix Fig S3E). Overcoming this issue by analyzing equivalent amounts of tissue for both genotypes, we determined that TH protein levels were higher in the BAT of *Cdk4^{-/-}* mice. Similarly, both the synaptic protein PSD95 and UCP1 were also increased in these conditions (Appendix Fig S3F and I).

In addition to its role in regulating BAT activity, sympathetic innervation can also stimulate the browning of WAT (Giralt & Villarroya, 2013). In line with the observed increase in iBAT sympathetic innervation, we found that *Cdk4^{-/-}* mice also exhibited browning of scWAT when they were kept at room temperature conditions. Indeed, hematoxylin–eosin staining showed the presence of clusters of adipocytes containing smaller and fragmented lipid droplets in the scWAT of *Cdk4^{-/-}*, compared to *Cdk4^{+/+}*, which is typical of beige adipocytes (Fig 4C). Moreover, we also found in the scWAT of *Cdk4^{-/-}* mice significantly increased expression of the *Adrb3* (β3-adrenergic receptor) gene (Fig 4D), but no impact in the content and subcellular localization of the protein ADRB3 was found (Appendix Fig S1A and C). We also noted a robust, yet not significant, trend toward increased expression of other thermogenic and beige genes, including *Dio2* and *Ucp1* (Fig 4D). Nevertheless, we have observed no differences in the sympathetic innervation in the scWAT of Cdk4$^{-/-}$ mice, as shown by the IHC analysis of TH (Appendix Fig S2A and B).

As thermogenic activation of brown adipocytes is mediated mainly by the β3-adrenergic receptor (ADRB3), we next treated the mice with a specific ADRB3 antagonist (SR59230A, called SR from now on) and analyzed their response to cold. As expected, *Cdk4^{+/+}* mice treated with the antagonist (SR) were more sensitive to cold exposure than vehicle-treated *Cdk4^{+/+}* mice (NaCl; Fig EV3A and B). Surprisingly, inhibition of β3-adrenergic signaling with SR did not decrease the body temperature in response to cold of the *Cdk4^{-/-}* mice, and SR-treated *Cdk4^{-/-}* mice exhibited the same cold resistance as NaCl-treated *Cdk4^{-/-}* mice (Fig EV3A and B). The activation of the ADRB3 by sympathetic stimulation leads to an increase in cAMP concentration inside the adipocyte and the subsequent activation of the PKA pathway that ultimately facilitates the release of stored fatty acids, providing substrates for oxidation and heat production. Western blot analysis of phosphorylated (activated) cAMP response-element binding protein (p-CREB^Ser133), which is a marker of activated PKA signaling, showed a robust phosphorylation of CREB^Ser133 in the iBAT of SR-treated *Cdk4^{-/-}* mice, whereas it was downregulated in the BAT of SR-treated *Cdk4^{+/+}* mice (Fig EV3C and D). These results suggest that

thermogenesis and CREB phosphorylation are not regulated by β3-adrenergic receptor signaling in *Cdk4^{-/-}* mice.

It has been well established that additional non-β3-adrenergic receptors also promote BAT activation (Susulic *et al*, 1995). Therefore, to explore the role of adrenergic signaling in general on thermogenesis in *Cdk4^{-/-}* mice, we used a one-time treatment of propranolol to inhibit all β-adrenergic signaling and measured response to a cold test. Propranolol did not affect the body temperature of *Cdk4^{+/+}* mice but significantly reduced the cold tolerance of *Cdk4^{-/-}* mice to levels even lower than those of the control group (Fig 4E and F). These results showed that the overactivation of BAT in *Cdk4^{-/-}* mice exposed to cold depends largely on increased β-adrenergic receptor signaling, but not on β3-adrenergic receptors specifically.

## SF1 neuron-specific deletion of *Cdk4* does not cause significant changes in body weight, body composition, oxygen consumption, or respiratory exchange rate

Since the VMH is one of the hypothalamic nuclei involved in the regulation of BAT thermogenesis (Lopez, *et al*, 2010), and as the observations in *Cdk4^{-/-}* mice (and supported by the normal phenotype of *Cdk4^flox/flox* Ucp1-Cre animals) suggested a key role for the CNS in their increased cold tolerance, we next explored the effects of the deletion of *Cdk4* in VMH SF1 neurons using *Cdk4^flox/flox* Sf1-Cre mice. We generated this mouse model by breeding *Cdk4^flox/flox* mice with transgenic mice that express Cre recombinase under the control of the steroidogenic factor-1 (Sf1) promoter/enhancer. To test the specificity of the deletion, we analyzed dissected VMH and observed a 50% reduction in *Cdk4* mRNA expression in *Cdk4^flox/flox* Sf1-Cre mice relative to control mice (Fig 5A). This represents a moderate yet significant decrease in VMH *Cdk4* levels, and the residual levels can be explained by the expression of *Cdk4* in other non-Sf1 VMH populations.

Control and *Cdk4^flox/flox* Sf1-Cre mice showed similar body weight (Fig EV4A). Body composition analysis using EchoMRI showed that *Cdk4^flox/flox* Sf1-Cre mice exhibited a modest decrease in their fat mass, but this decrease did not reach statistical significance (Fig EV4B). No differences were found in VO$_2$ (Fig EV4C and D), RER (Fig EV4E and F), and EE (Fig EV4G and H) between the two groups at room temperature. Similarly, when the energy balance was calculated, no differences were observed between *Cdk4^flox/flox* Sf1-Cre mice and their control littermates at room temperature (Fig EV4I and J). These data suggested that the deletion of Cdk4 in SF1 neurons is not sufficient to alter whole-body energy metabolism at least under the conditions tested.

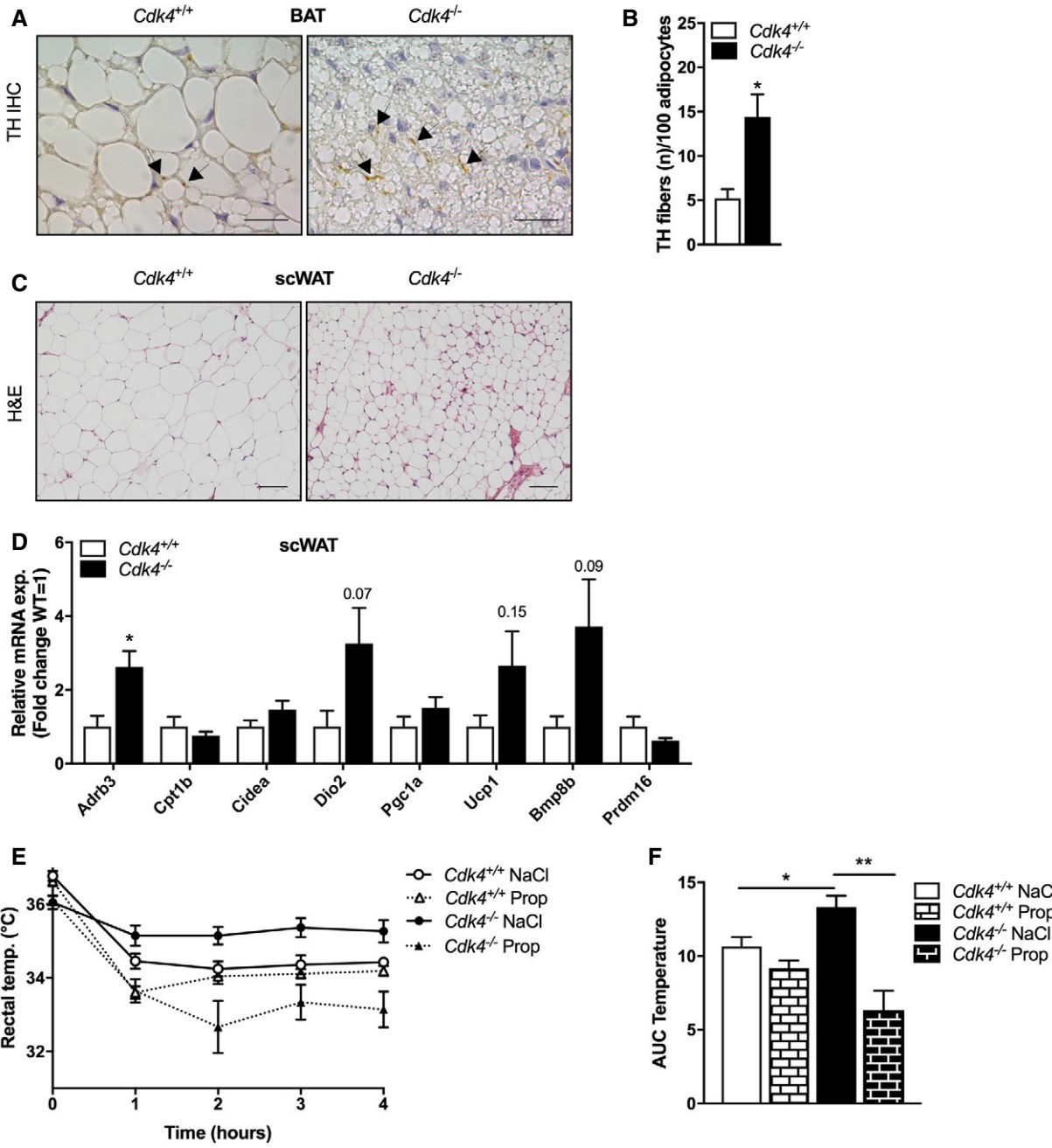

**Figure 4. Activation of thermogenesis by increased sympathetic innervation of BAT in $Cdk4^{-/-}$ mice is blunted by β-adrenergic antagonist treatment.**

A, B   TH immunohistochemical staining in iBAT sections (A) (scale bar 20 μm, arrows indicate TH parenchymal fibers) and corresponding quantification of the number of TH fibers relative to 100 adipocytes (B) ($Cdk4^{+/+}$ [n = 6] and $Cdk4^{-/-}$ [n = 6]).

C   Hematoxylin–eosin staining in scWAT showing multilocular adipocytes in $Cdk4^{-/-}$ (scale bar 50 μm).

D   Expression of thermogenic genes and brown adipocyte markers in scWAT of $Cdk4^{+/+}$ (n = 5) and $Cdk4^{-/-}$ (n = 6) mice as assessed by RT–PCR.

E, F   Acute cold test (4°C) after treatment with the β-adrenergic antagonist propranolol (Prop) in $Cdk4^{+/+}$ (n = 7) and $Cdk4^{-/-}$ (n = 6) animals or with vehicle (NaCl; $Cdk4^{+/+}$ [n = 7] and $Cdk4^{-/-}$ [n = 6]) (E) and corresponding quantification of the area under the curve (AUC).

Data information: All data are shown as the mean ± SEM; Student's t-test was used for statistical analysis. *P < 0.05, **P < 0.01.
Source data are available online for this figure.

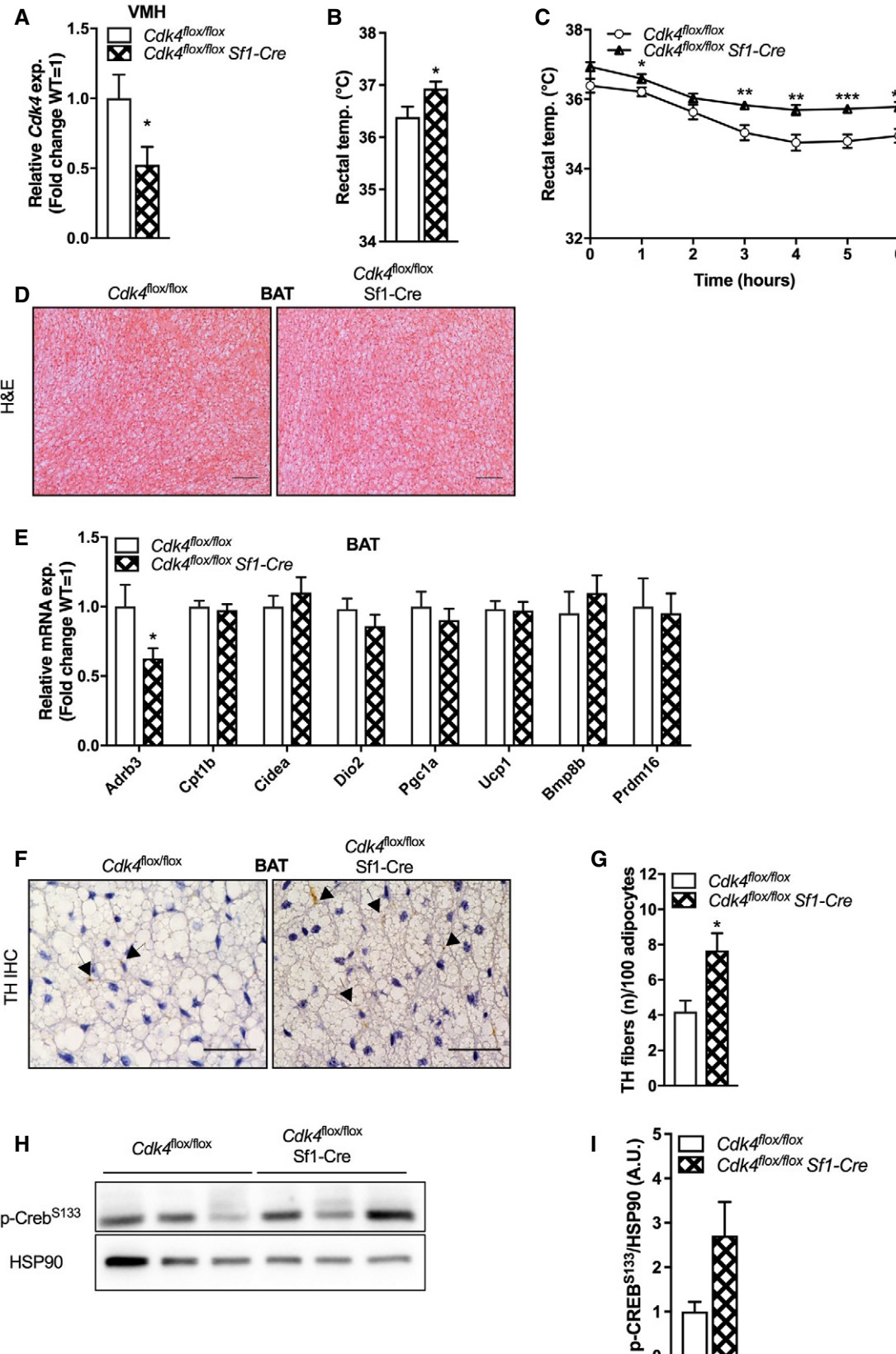

**Figure 5.**

**Figure 5. *Cdk4* deficiency in VMH SF1 neurons leads to increased heat production during cold exposure.**

A    *Cdk4* mRNA expression in the dissected VMH of $Cdk4^{flox/flox}$ ($n = 5$) and $Cdk4^{flox/flox}$ Sf1-Cre ($n = 6$) mice.

B, C    Rectal temperature was measured at 24°C (B), and rectal temperature was monitored during acute cold exposure at 4°C for 6 h (C) in $Cdk4^{flox/flox}$ ($n = 15$) and $Cdk4^{flox/flox}$ Sf1-Cre mice ($n = 15$).

D    Hematoxylin–eosin (H&E) staining of iBAT sections (scale bar 100 μm) from $Cdk4^{flox/flox}$ and $Cdk4^{flox/flox}$ Sf1-Cre mice.

E    Expression of thermogenic genes and brown adipocyte markers expression in iBAT $Cdk4^{flox/flox}$ ($n = 9$) and $Cdk4^{flox/flox}$ Sf1-Cre mice ($n = 9$) mice as assessed by RT–PCR.

F, G    TH immunohistochemical (IHC) staining in iBAT sections (F) (scale bar 40 μm, arrows indicate TH parenchymal fibers) and corresponding quantification of the number of TH fibers relative to 100 adipocytes (G) ($Cdk4^{flox/flox}$ [$n = 5$] and $Cdk4^{flox/flox}$ Sf1-Cre mice [$n = 5$]).

H, I    Western blot analysis (H) and quantification of p-CREB$^{S133}$ protein expression following 6 h of cold exposure (4°C) (I) ($n = 3$ biological replicates). HSP90 was used as a loading control.

Data information: All data are shown as the mean ± SEM. Student's *t*-test (A, B, C, E, G) and Mann–Whitney *U*-test (I) were used for statistical analyses.

*$P < 0.05$, **$P < 0.01$, ***$P < 0.001$.

Source data are available online for this figure.

## Increased cold tolerance and sympathetic input to BAT and scWAT in $Cdk4^{flox/flox}$ Sf1-Cre mice

To explore whether the phenotype observed in the $Cdk4^{-/-}$ mice may be due to the lack of *Cdk4* in SF1 neurons specifically, we examined the cold tolerance of the $Cdk4^{flox/flox}$ Sf1-Cre mice subjected to a cold test at 4°C for 6 h. $Cdk4^{flox/flox}$ Sf1-Cre mice showed a significant increase in basal body temperature (Fig 5B) and a robust increase in cold resistance during the cold test when compared to control littermates (Fig 5C). No morphological differences were observed in hematoxylin–eosin-stained iBAT between both groups (Fig 5D), and thermogenic gene expression in iBAT was also unaffected (Fig 5E). Interestingly, we observed a higher number of TH-positive parenchymal fibers in the iBAT of $Cdk4^{flox/flox}$ Sf1-Cre mice relative to controls (Fig 5F and G), reminiscent of the observations in $Cdk4^{-/-}$ mice also validated by western blotting (Appendix Fig S4A and B). Protein levels of p-HSL S660 were also analyzed in these animals revealing no differences in its phosphorylation between genotypes at room temperature conditions (Appendix Fig S4C and D). However, western blot analysis showed increased levels of CREB phosphorylation in the iBAT of $Cdk4^{flox/flox}$ Sf1-Cre mice after acute exposure to cold (Fig 5H and I).

Another tissue that is also responsive to cold and is regulated by the brain is the scWAT. Therefore, we also explored the appearance and the molecular phenotype of the scWAT of $Cdk4^{flox/flox}$ Sf1-Cre mice. Gene expression analysis of thermogenic markers indicated an increased expression of *Prdm16*, a key regulator of beige adipocyte recruitment and thermogenic function in scWAT of $Cdk4^{flox/flox}$ Sf1-Cre mice (Fig EV5A). Consistently with this observation, analysis of sympathetic innervation by immunohistochemical analysis of TH showed an increased amount of TH protein expression (Fig EV5B and C), which was further confirmed by western blotting (Fig EV5D and E). Moreover, deletion of *Cdk4* in SF1 neurons resulted in increased UCP1 expression in scWAT under basal conditions (Fig EV5D and F).

Taken together, these data suggest that deletion of *Cdk4* in VMH SF1 neurons increases sympathetic innervation of BAT and scWAT, which in turn leads to increased thermogenesis during exposure to cold.

## Increased c-Fos-immunoreactive cells in specific hypothalamic nuclei involved in thermoregulation in $Cdk4^{-/-}$ mice

Despite being cold-resistant and having increased BAT innervation, $Cdk4^{flox/flox}$ Sf1-Cre mice do not fully phenocopy the characteristics of the BAT of the $Cdk4^{-/-}$ mice, suggesting that other mechanisms could also participate in the regulation of the BAT function in these mice. Thermogenesis is tightly regulated by the central nervous system. Several brain regions have been described to command the responses to thermal stress, including the medial preoptic area (mPOA), the arcuate nucleus of the hypothalamus (ARC), the VMH, the dorsomedial hypothalamus (DMH), and the lateral hypothalamic area (LH). We therefore investigated the impact of *Cdk4* deletion in hypothalamic neural activation (c-Fos-immunoreactive [IR] cells) in $Cdk4^{+/+}$ and $Cdk4^{-/-}$ mice at room temperature (Fig 6A–D). Quantification of c-Fos-immunoreactive cells in the brain sections showed a significant increase in c-Fos-IR cells in the ARC, VMH, and LH; and a clear tendency in the mPOA, in the brain sections of $Cdk4^{-/-}$ compared to $Cdk4^{+/+}$ control mice (Fig 6E). These observations indicated that the enhanced BAT activation found in the $Cdk4^{-/-}$ mice results from the coordination of increased activation of several thermoregulatory regions in the brain of these animals.

## Discussion

The cell cycle regulator CDK4, in addition to its well-established role in the control of cell cycle, is also known to be involved in multiple aspects of metabolism, including a positive role in adipogenesis, through the regulation of the activity of PPARγ, and the control of

**Figure 6.    Activation of thermoregulatory brain regions in $Cdk4^{-/-}$ mice.**

A–D    Confocal illustrations of frontal hypothalamic sections from $Cdk4^{+/+}$ mice (A, C, $n = 3$) and $Cdk4^{-/-}$ mice (B, D, $n = 3$). Low magnification showing nuclei (Hoechst, left panel) and c-Fos-activated cells (white, right panel) in the mPOA (A1–2, B1–2) and the ARC, the VMH, the DMH, the LH (C1–2, D1–2). High magnifications corresponding to the frame area in (A1–2), (B1–2), (C2), (D2) and showing c-Fos-IR cells (A3–4, B3–4, C3–6, D3–6). Scale bars (A1–2, B1–2, C1–2, D1–2): 200 μm, (A3–4, B3–4, C3–4, D3–4): 50 μm.

E    Quantification of c-Fos IR cells in brain sections of $Cdk4^{+/+}$ and $Cdk4^{-/-}$ mice ($n = 15$–24 slices according to hypothalamic nucleus and genotype).

Data information: All data are shown as the mean ± SEM; Mann–Whitney *U*-test was used for statistical analysis. **$P < 0.01$.

Source data are available online for this figure.

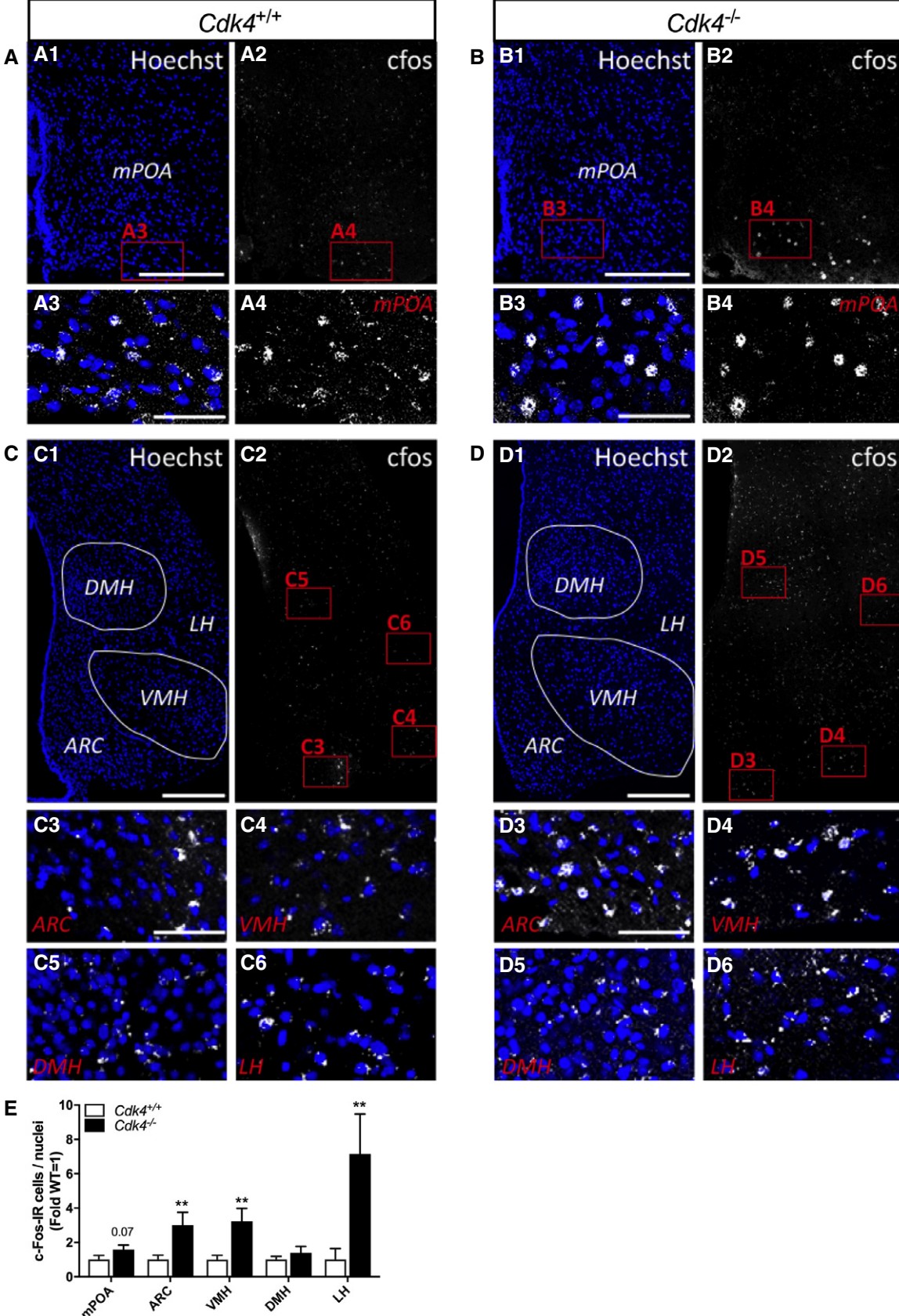

**Figure 6.**

the insulin signaling pathway, via the phosphorylation of IRS2 (Abella *et al*, 2005; Lagarrigue *et al*, 2016; Iqbal *et al*, 2018). In addition, CDK4 directly modulates the oxidative metabolism through the inhibition of the activity of AMPK by direct phosphorylation (Lopez-Mejia *et al*, 2017). Since the thermogenic function of BAT is also dependent on oxidative processes and mitochondrial respiration, we tested in this study whether $Cdk4^{-/-}$ mice, similar to the increased fatty acid oxidation that we observed in muscle, also had increased oxidative capacity in BAT. We show here that $Cdk4^{-/-}$ mice have increased EE and oxygen consumption, especially during cold exposure, accompanied by a metabolic switch toward lipid utilization. Moreover, $Cdk4^{-/-}$ mice showed improved maintenance of body temperature in response to cold and have increased expression of *Ucp1* and other thermogenic genes, including *Cidea* and *Dio2*, in BAT. To our surprise, however, we found that the increased thermogenesis in the $Cdk4^{-/-}$ mice was not due to a direct function of CDK4 in the BAT, since none of these changes were observed in the BAT-specific *Cdk4* knockout ($Cdk4^{flox/flox}$ Ucp1-Cre) mice.

The activity of BAT is tightly regulated by the sympathetic nervous system (SNS) (Morrison *et al*, 2014; Contreras *et al*, 2015). Indeed, the iBAT of $Cdk4^{-/-}$ mice was highly innervated by the SNS, as measured by the presence of TH-positive fibers, indicating that at baseline, CDK4 downregulates BAT activity by attenuating sympathetic innervation. To investigate whether β-adrenergic receptor signaling in general mediated the observed alterations in thermogenesis in $Cdk4^{-/-}$ mice, we treated them with propranolol, a broad β-adrenergic receptor antagonist. When treated with propranolol, $Cdk4^{-/-}$ mice showed increased sensitivity to cold, as evidenced by impaired thermogenesis during cold exposure. In fact, their body temperature decreased below that of control animals, thus indicating that the higher BAT activation observed in $Cdk4^{-/-}$ mice can be explained by a pan-β-adrenergic receptor activation.

The VMH is a hypothalamic nucleus that is highly enriched in SF1 neurons and is known to regulate BAT activation (Kim *et al*, 2011; Martinez de Morentin *et al*, 2014; Zhang & Bi, 2015). Deleting *Sf1* in the brain increases the cold sensitivity of mice by decreasing BAT-mediated thermogenesis (Kim *et al*, 2011). Recent studies have suggested that the central control of BAT activity is mediated by AMP-activated protein kinase (AMPK) in the SF1 neurons of the VMH (Seoane-Collazo *et al*, 2018). Other studies have also shown that thyroid hormones (Martinez de Morentin *et al*, 2014; Martinez-Sanchez *et al*, 2017), estradiol (Lopez *et al*, 2016), and bone morphogenetic proteins (BMPs) regulate the function of brown adipocytes (Whittle *et al*, 2012; Ghaben & Scherer, 2019). Indeed, the adipokine BMP8B, thyroid hormones, and estradiol are involved in the neuroregulation of non-shivering thermogenesis at the level of the VMH, where they decrease AMPK activity to increase BAT activation (Lopez *et al*, 2010; Martins *et al*, 2016). Taken together, these studies suggest that the VMH is an especially important nucleus for the regulation of brown fat activity. To analyze whether CDK4 could participate in the control of the thermogenic function of VMH, we generated SF1 neuron-specific *Cdk4* knockout mice ($Cdk4^{flox/flox}$ Sf1-Cre). Strikingly, the thermogenic phenotype in these mice was the same as that observed in $Cdk4^{-/-}$ mice, with decreased sensitivity to cold, demonstrating that CDK4 in the VMH contributes to the regulation of cold-induced thermogenesis. In contrast to the $Cdk4^{-/-}$ mice, the $Cdk4^{flox/flox}$ Sf1-Cre mice did not

have increased expression of thermogenic markers in the iBAT under basal conditions. We can speculate that these differences may arise from the role of CDK4 in other regions of the hypothalamus. Neuronal activity in response to cold is induced in specific areas of the hypothalamus, including mPOA, DMH, VMH, LH, and ARC. Here, we show indeed that the activation of neurons is increased in the ARC, LH, and VMH of the $Cdk4^{-/-}$ mice. The ARC contains orexigenic (AgRP, NPY) and anorexigenic (POMC, CART) neuronal populations controlling feeding behavior which can also modulate BAT activity (Yasuda *et al*, 2004; Shi *et al*, 2013). Interestingly, $Cdk4^{-/-}$ mice exhibited both increased thermogenesis and increased food intake, suggesting that the ARC is actually constitutively activated in $Cdk4^{-/-}$ mice. The VMH has a very important role in the regulation of the thermogenic response. Numerous authors have postulated a neuronal control of BAT function mediated by the VMH. This nucleus integrates several signals; among them, the thermogenic factor BMP8B signaling, which coordinates sympathetic firing in BAT through the inhibition of AMPK in the VMH and the activation of orexin signaling in the LH, coordinates sympathetic firing in BAT (Martins *et al*, 2016). The participation of CDK4 in these other areas of the hypothalamus, like the ARC and the LH, remains to be investigated.

Another important tissue contributing to thermogenesis is the scWAT. Upon sustained thermal stress, brown-like adipocytes arise in scWAT and can contribute to heat generation in a process called browning (Bartelt & Heeren, 2014). Our findings in $Cdk4^{flox/flox}$ Sf1-Cre mice demonstrate an induction of browning in scWAT, as shown by the enhanced expression of UCP1 and thermogenic genes. These results are also supported by an increased TH presence. Therefore, CDK4 in VMH impacts thermogenesis as a result of the combined effects in sympathetic innervation in BAT and scWAT.

The results of our study go beyond a specific function of CDK4 in the inhibition of oxidative processes in specific cells and tissues. We prove that CDK4 activity is important for the regulation of energy homeostasis at the whole organism level. Through its action at the VMH, and likely in other regions of the hypothalamus, CDK4 controls the activity of a peripheral tissue, in this case the BAT. The relevance of this finding for the function of other tissues remains to be analyzed. It will also be important to decipher the exact molecular mechanisms by which CDK4 controls the function of the VMH, and of the hypothalamus in general. Indeed, CDK4 obviously participates in the regulation of the growth of the organism: mice deficient in CDK4 are smaller (Rane *et al*, 1999). This global function of CDK4 that we describe in the present study would be in complement to the cellular and tissue control of metabolism and cell cycle progression by CDK4. Altogether, the general function of CDK4 is to redirect energy availability to biosynthetic processes, rather than to energy-consuming processes like heat generation and dissipation, resulting for example in a decrease in BAT activation. This is consistent with the known role of CDK4 in controlling several biosynthetic pathways (Lagarrigue *et al*, 2016; Lopez-Mejia *et al*, 2017).

The study of BAT thermogenesis has raised much interest in recent years since BAT activation increases energy expenditure and contributes to the maintenance of glucose homeostasis (Lowell *et al*, 1993; Cannon & Nedergaard, 2009; Stanford *et al*, 2013; Chondronikola *et al*, 2014). These traits make BAT a tissue of interest for the design of targeted therapies against obesity and diabetes (Cypess & Kahn, 2010; Trayhurn, 2016; Soler-Vazquez *et al*, 2018). In

addition to elucidating a new role of CDK4 in the control of energy homeostasis, the effects of CDK4 depletion in BAT function justify to set up new studies based on the use CDK4 inhibitors for the treatment of metabolic diseases.

# Materials and Methods

## Animals

The generation of $Cdk4^{-/-}$ ($Cdk4^{nc/nc}$) mice has been previously described (Lagarrigue et al, 2016). Ucp1-Cre (Jackson Laboratories, Bar Harbor, ME) and Sf1-Cre mice (Jackson Laboratories, Bar Harbor, ME, USA) on a C57BL6/J background were intercrossed with $Cdk4^{flox/flox}$ (generated in collaboration with Cyagen Biosciences, Santa Clara, CA, USA) to generate the two experimental models used in this study ($Cdk4^{flox/flox}$ Ucp1-Cre and $Cdk4^{flox/flox}$ Sf1-Cre), and genotype was validated by PCR. Animals were maintained in a temperature-controlled animal facility with a 12-h light/12-h dark cycle and free access to standard chow diet and water according to the Swiss Animal Protection Ordinance (OPAn). Only male animals were used in this study. Body composition (fat and lean mass) was measured using EchoMRI technology with the EchoMRI™ qNMR system (Echo Medical Systems, Houston, TX, USA) under isoflurane anesthesia at the Mouse Metabolic Evaluation Facility (MEF) of the University of Lausanne. Oxygen consumption ($VO_2$) and the RER were measured with an Oxymax/CLAMS system (Columbus Instruments, Columbus, OH, USA) on a 12-h light/12-h dark cycle and free access to standard chow diet and water according to the Swiss Animal Protection Ordinance (OPAn) at 24°C. For the measurements done in cold, temperature of the room was changed to 6°C. $VO_2$ values were normalized to body weight. Energy expenditure was calculated as previously described in Ref. (Nie et al, 2015) using the following formula: $EE = ((RER \times 1.232) + 3.815) \times VO_2)/1,000$. In addition, the Cal R program (Mina et al, 2018) was used to implement analysis of a generalized linear model (GLM) analysis utilizing body mass as a covariate when modeling mass-dependent metabolic parameters using a "two-group" template. The results of this analysis are depicted in a table, where the significance test for "Group effect" is whether the two groups of interest are significantly different for the metabolic variable selected and the significance test for "Mass_effect_" determines whether there is an association between the mass variable and metabolic variable selected among all animals in the study. Regression plots for energy balance (calculated considering the caloric value of the diet used: Kliba-Nafag 3436 of 3.12888125 Kcal/g). An acute cold test was carried out on animals individually housed at 4°C for 6 h, and rectal temperature was measured every hour using a thermal probe (Bioseb Lab Instruments, Vitrolles, France). Unless stated otherwise, animals were sacrificed at room temperature, after an overnight fasting, by cervical dislocation, and iBAT, scWAT, and brain were collected. For the experiments on $Cdk4^{flox/flox}$ Sf1-Cre mice, VMH was dissected from frozen sections of the brain. BAT was weighed after dissection. Then, a tissue sample was used for histological analysis, and the remaining tissue was directly snap-frozen in liquid nitrogen. All animal care and treatment procedures were performed in accordance with the Swiss guidelines and were approved by the Canton of Vaud SCAV (authorization VD 3121.h).

## β-Adrenergic receptor antagonist experiments

For the β3-antagonist experiment, SR-59230A (S8688, Sigma) was dissolved in saline and injected i.p. at 3 mg/kg/day for 7 days (Lopez et al, 2010). An acute cold test (4°C) was performed on day 5, where body temperature was monitored every hour for 6 h using a thermal probe (Bioseb Lab Instruments, Vitrolles, France). To broadly inhibit β-adrenergic receptors, propranolol (Sigma) was dissolved in saline and injected i.p. at 5 mg/kg once (Banfi et al, 2018). Two hours later, mice were subjected to an acute cold test (4°C) where body temperature was monitored every hour for a total of 4 h using a thermal probe (Bioseb Lab Instruments, Vitrolles, France).

## Respirometry measurements

Fresh BAT was homogenized immediately after dissection to assess respiration using high-resolution respirometry (Oroboros Oxygraph-2k, Oroboros Instruments, Innsbruck, Austria). Tissue homogenization was performed by mechanical permeabilization in MiR05 medium (0.5 mM EGTA, 3 mM $MgCl_2$, 60 mM lactobionic acid, 20 mM taurine, 10 mM $KH_2PO_4$, 20 mM HEPES, 110 mM D-sucrose, 1 g/l BSA, pH 7.1), and the equivalent of 2 mg of tissue was added to the experimental respirometry chamber. Malate (2 mM), pyruvate (10 mM), and glutamate (20 mM) were first added to the chamber without ADP to measure oxygen flux (called "leak" in Fig 2H). Complex-I-dependent respiration ("CI", in Fig 2H). was next measured after adding ADP (5 mM). Then, succinate (10 mM) was added to analyze electron flow through both complexes I and II ("CI + CII", in Fig 2H). This was followed by the addition of carbonylcyanide-4-(trifluoromethoxy)-phenyl-hydrazone (FCCP) to obtain maximum flux through the electron transfer system ("ETS," in Fig 2H). Finally, oxidative phosphorylation was fully inhibited by the addition of rotenone (0.1 μM) and antimycin A (2.5 μM; "ETS (CII)", in Fig 2H). The remaining $O_2$ flux resulting from the inhibition of antimycin A (ETS-independent flux) was subtracted from the values obtained from each of the previous measurements. $O_2$ flux values are expressed relative to leak respiration.

## Histology

Hematoxylin–eosin staining was performed on 4-μm-thick formalin-fixed paraffin-embedded iBAT and scWAT sections to evaluate tissue morphology. Immunohistochemical analyses of UCP1 and TH expression were performed using the antibodies anti-UCP1 (ab10983, Abcam) and anti-TH (AB1542, Merck Millipore). TH expression was analyzed by counting the number of TH-positive parenchymal fibers in 10 high magnification (60×) fields of each iBAT and scWAT section. For each animal of a total of five per group, three or four representative sections collected every 200 μm were analyzed. Data are expressed as the number of TH-positive fibers to 100 adipocytes. Morphometric analyses were performed by IS, WB, and AG.

## Mitochondria and lipid droplet quantification

Mitochondria and lipid droplet volumes were quantified using electromicrographs from BAT of $Cdk4^{-/-}$ and $Cdk4^{+/+}$ mice. In brief, samples were fixed, embedded, and cut in ultrathin sections of 50 nm with a Leica Ultracut (Leica Mikrosysteme GmbH, Vienna, Austria). Four micrographs with pixel size of 6.86 nm were taken with a transmission electron microscope Philips CM100 (Thermo Fisher Scientific, Waltham, MA, USA) at an acceleration voltage of 80 kV with a TVIPS TemCam-F416 digital camera (TVIPS GmbH, Gauting, Germany) and assembled to form a final image that was analyzed using the program IMOD (IMOD 4.7, Boulder Laboratory for 3-D Electron Microscopy of Cells, CO, USA) to determine mitochondrial and lipid volume. A grid of 500 × 500 nm was superimposed on each micrograph. Then, to assess mitochondrial and lipid volume, the points (defined as the intersection of two grid lines) that were on a mitochondria or lipid droplet were counted and divided by the total amount of points that were on cytoplasmic space.

## RNA extraction and RT–PCR

scWAT and iBAT were powdered manually in liquid nitrogen. RNA was isolated with Tri-Reagent (T9424, Sigma-Aldrich). For scWAT and BAT, 500 µl of Tri-Reagent was used to lyse 30 mg of tissue powder. After centrifugation to remove debris (12,000 *g*, 10 min, 4°C), 100 µl of chloroform was used for phase separation. The aqueous phase was recovered and precipitated with isopropanol, and centrifuged (12,000 *g*, 10 min, 4°C), and then, RNA pellets were washed with 500 µl of 75% ethanol. Pellets were resuspended in 200 µl of Milli-Q water. A second step of 1:1 chloroform was used to obtain better-purified RNA. The aqueous phase was recovered and incubated overnight at −20°C with 70 µl of ammonium acetate (NH$_4$AC) and 600 µl of absolute ethanol. Then, precipitation of RNA pellets was performed by centrifugation (12,000 *g*, 30 min, 4°C). Pellets were washed with 75% ethanol followed by centrifugation (12,000 *g*, 10 min, 4°C) and were resuspended in Milli-Q water. RNA concentrations were determined using NanoDrop and reverse-transcribed using 1,000 ng of RNA and Superscript II enzyme (18064014, Invitrogen) according to the manufacturer's instructions. qPCR analysis was performed using SYBR Green detection (04913914001, Roche) on a 7900HT Fast Real-Time PCR System (Applied Biosystems) according to the manufacturer's instructions. Relative mRNA expression was calculated from the comparative threshold cycle (Ct) values of the gene of interest relative to RS9 mRNA. The specific primer sequences that were used are as follows: *Adrb3* (F′-CACCGCTCAACAGGTTTGATG, R′-TCTTGGGGCAAC-CAGTCAAG), *Bmp8b* (F′-TCCACCAACCACGCCACTAT, R′-CAGTAG GCACACAGCACACCT), *Cdk4* (F′-GCCTGTGTCTATGGTCTG, R′-AAGCAGGGGATCTTACGC), *Cidea* (F′-TGCTCTTCTGTATCGCCCA GT, R′-GCCGTGTTAAGGAATCTGCTG), *Cpt1b* (F′-CCGGAAAGGTA TGGCCACTT, R′-GAAGAAAATGCCTGTCGCCC), *Dio2* (F′-CAAA-CAGGTTAAACTGGGTGAAGA, R′-GTCAAGAAGGTGGCATTCGG), *Pgc1α* (F′-CCGATCACCATATTCCAGGTC, Prdm16 (F′-CAGCACGGT-GAAGCCATTC, R′-GCGTGCATCCGCTTGTG), *Rs9* (F′-C GGCCCGGGAGCTGTTGACG, R′-CTGCTTGCGGACCCTAATGTGAC G), and *Ucp1* (F′-CACCTTCCCGCTGGACACTGC, R′-TTGCCAGGGT GGTGATGGTCC).

## Protein extraction and western blot analysis

Proteins from BAT, scWAT, and pgWAT were extracted using mammalian protein extraction reagent (MPER, Pierce) supplemented with halt phosphatase and halt protease inhibitors (Pierce) as per the manufacturer's instructions, separated by gel electrophoresis and analyzed using the antibodies anti-CDK4 (C-22, sc-260, Santa Cruz), anti-p-CREB Ser133 (87G3, 9198S, Cell Signaling Technology), anti-CREB total (48H2, 9197S, Cell Signaling Technology), anti-α-tubulin (DM1A, Sigma), anti-UCP1 (ab10983, Abcam), anti-HSP90 (4874S, Cell Signaling Technology), anti-GAPDH (6C5, sc-32233, Santa Cruz), and anti-TH (AB154, Millipore).

## c-Fos immunofluorescence

Sample preparation and immunofluorescence labelings were performed essentially as previously described (Geller *et al*, 2017, 2019). Mice were deeply anesthetized with intraperitoneal administration of sodium pentobarbital (100 mg/kg). When intercostal muscle paralysis occurred, the thoracic cavity was opened, and the left ventricle was perfused with 15 ml of heparine sodium salt (54 mg/l, Sigma H9399) in PBS followed by 50 ml of 4% paraformaldehyde (PAF, Sigma 158127) in PBS (pH 7.4). The brain was then removed, post-fixed during 1 h in PFA 4% at 4°C, and cryoprotected in 30% sucrose at 4°C during 48 h. Brains were embedded in Tissue-Tek® O.C.T™compound (Sakura #4583) and frozen in carbonic ice before sectioning. Four sets of coronal sections (20 µm thick) from the preoptic area to the premammillary nucleus were cut on a cryostat (Thermo Fisher Scientific, NX50) and directly mounted on a Superfrost Plus slide (Thermo Fisher Scientific). Coronal sections of $Cdk4^{+/+}$ (n = 3) and $Cdk4^{-/-}$ (n = 3) brain were incubated for 20 min in blocking solution (2% donkey serum albumin in PBS 0.3% Triton X-100) at room temperature (RT) and incubated overnight at 4°C with rabbit anti-cfos (1:500, Cell Signaling #22505 9F6). Tissue was then rinsed three times in PBS and incubated in secondary antibody donkey anti-rabbit IgG (H+L) Alexa Fluor® 568 (1:500, Life Technology A10042) for 2 h and 30 min at RT. Nucleus staining was performed by incubation with Hoechst 33342 solution (Life Technology H21492) for 2 min at RT.

## Image acquisition and quantification analysis

Confocal images were acquired with a laser scanning microscope, Zeiss LSM 880, and the associated ZEN software using tile scan mode (512 × 512 mosaics, Pinhol 9) and 40× objective. mPOA was analyzed from Bregma 0.74 to 0.26 mm (n = 19 slices $Cdk4^{+/+}$ mice, n = 24 slices $Cdk4^{-/-}$ mice); ARC from Bregma −1.22 to −2.06 mm (n = 22 slices $Cdk4^{+/+}$ mice, n = 20 slices for $Cdk4^{-/-}$ mice); VMH from Bregma −1.06 to −2.06 mm (n = 20 slices $Cdk4^{+/+}$ mice, n = 21 slices $Cdk4^{-/-}$ mice); DMH from Bregma −1.46 to −2.06 mm (n = 17 slices $Cdk4^{+/+}$ mice, n = 15 slices $Cdk4^{-/-}$ mice); and LH from Bregma −1.06 to −1.70 mm (n = 15 slices $Cdk4^{+/+}$ mice, n = 16 slices $Cdk4^{-/-}$ mice). Quantification of number of nuclei (Hoechst) and of c-Fos-IR cells was performed using ImageJ2 Software (Rueden *et al*, 2017) using the "Analyze particles" mode (size µm$^2$: 20–200) in each hypothalamic area/nuclei predefined manually in each slice and for each animal. The number of c-Fos-IR cells was reported on the number of nuclei

detected in each hypothalamic area/nucleus for each slice from each mouse (which corresponds to the percentage of c-Fos-IR cells). The values are represented in fold change of percentage of c-Fos-IR cells of $Cdk4^{-/-}$ slices compared to $Cdk4^{+/+}$ slices.

### Statistical analysis

Data are presented as the mean $\pm$ SEM. Data distribution was analyzed by the Shapiro–Wilk normality test. Statistical analyses were carried out with Student's $t$-tests (normal distribution) and Mann–Whitney $U$-test. Differences were considered statistically significant at *$P$-value < 0.05, **$P$-value < 0.01, ***$P$-value < 0.001.

# Data availability

No data were deposited in a public database.

**Expanded View** for this article is available online.

## Acknowledgements

The members of the Fajas laboratory are acknowledged for support and discussions. We thank M. Barbacid for providing the $Cdk4^{-/-}$ mice. We thank Anne-Catherine Thomas, Brigitte Delacuisine, Manon Gervais, and CIG SSC for technical support. We thank Jean Daraspe and the Electron Microscopy Facility at University of Lausanne for the transmission electron microscopy photographs and for their advice in the quantification. This study was supported by the Swiss National Science Foundation (31003A_143369) and by Ministerio de Ciencia, Innovación y Universidades, co-funded by the FEDER Program of EU (ML: RTI2018-101840-B-I00). ICLM is supported by the Swiss National Science Foundation (Ambizione PZ00P3_168077).

## Author contributions

JC-A and ICL-M designed studies, conducted experiments, analyzed data, and participated in writing. VB, SG, HJ, IS, WV, EAF, CM, KH, LCL-E, AN, LM-C, GN, PS-C, SL, and SC conducted experiments. ML, AG, and BT designed studies and provided reagents. LF designed studies, analyzed data, and participated in writing.

## Conflict of interest

The authors declare that they have no conflict of interest.

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
