## [Review Process File · EMBO Reports]

Hypothalamic CDK4 regulates thermogenesis by modulating sympathetic innervation of adipose tissues

Judit Castillo-Armengol, Valentin Burquissau, Sarah Geller, Honglei Ji, Ilenia Severi, Wiebe Venema, Eric-Aria Fernandez, Catherine Moret, Katharina Huber, Lucia Leal-Esteban, Anita Nasrallah, Laia Martinez-Carreres, Guy Niederhäuser, Patricia Seoane Collazo, Sylviane Lagarrigue, Miguel Lopez, Antonio Giordano, Sophie Croisier, Bernard Thorens, Isabel Lopez-Mejia, and Lluís Fajas

DOI: [10.15252/embr.201949807](https://doi.org/10.15252/embr.201949807)

Corresponding author(s): *Lluís Fajas (lluis.fajas@unil.ch)*

Review Timeline:

Submission Date:	4th Dec 19
Editorial Decision:	8th Jan 20
Revision Received:	9th Apr 20
Editorial Decision:	12th May 20
Revision Received:	14th May 20
Accepted:	28th May 20

Editor: *Deniz Senyilmaz Tiebe*

Transaction Report:

Dear Lluís,

Thank you for the submission of your research manuscript to our journal, which was now seen by two referees, whose reports are copied below.

As you can see, the referees express interest in the analysis. However, they also raise a number of concerns that need to be addressed to consider publication here. In particular, referees require better support for beta-3-receptor activation (ref #1, point 2), better characterization/quantification of TH levels of CDK4 depleted mice (ref #1 points 3, 4 and ref #2 point 3) and calorimetry data (ref #2 point 1), and more support into neuronal specificity of CDK4 role (ref #2 point 4). I find the reports informed and constructive, and believe that addressing the concerns raised will significantly strengthen the manuscript.

Given these constructive comments, we would like to invite you to revise your manuscript with the understanding that the referee concerns (as in their reports) must be fully addressed and their suggestions taken on board. Please address all referee concerns in a complete point-by-point response. Acceptance of the manuscript will depend on a positive outcome of a second round of review. It is EMBO reports policy to allow a single round of revision only and acceptance or rejection of the manuscript will therefore depend on the completeness of your responses included in the next, final version of the manuscript.

1. A data availability section providing access to data deposited in public databases is missing (where applicable).
2. Your manuscript contains statistics and error bars based on $n=2$ or on technical replicates. Please use scatter plots in these cases.

Supplementary/additional data: The Expanded View format, which will be displayed in the main HTML of the paper in a collapsible format, has replaced the Supplementary information. You can submit up to 5 images as Expanded View. Please follow the nomenclature Figure EV1, Figure EV2 etc. The figure legend for these should be included in the main manuscript document file in a section called Expanded View Figure Legends after the main Figure Legends section. Additional Supplementary material should be supplied as a single pdf labeled Appendix. The Appendix includes a table of content on the first page with page numbers, all figures and their legends. Please follow the nomenclature Appendix Figure Sx throughout the text and also label the figures according to this nomenclature. For more details please refer to our guide to authors.

2) individual production quality figure files as .eps, .tif, .jpg (one file per figure).

3) a .docx formatted letter INCLUDING the reviewers' reports and your detailed point-by-point responses to their comments. As part of the EMBO Press transparent editorial process, the point-by-point response is part of the Review Process File (RPF), which will be published alongside your paper. For more details on our Transparent Editorial Process, please visit our website: <https://www.embopress.org/page/journal/14693178/authorguide#transparentprocess>
You are able to opt out of this by letting the editorial office know (emboreports@embo.org). If you do opt out, the Review Process File link will point to the following statement: "No Review Process File is available with this article, as the authors have chosen not to make the review process public in this case."

4) a complete author checklist, which you can download from our author guidelines (). Please insert information in the checklist that is also reflected in the manuscript. The completed author checklist will also be part of the RPF.

5) Please note that all corresponding authors are required to supply an ORCID ID for their name upon submission of a revised manuscript (). Please find instructions on how to link your ORCID ID to your account in our manuscript tracking system in our Author guidelines ().

6) We replaced Supplementary Information with Expanded View (EV) Figures and Tables that are collapsible/expandable online. A maximum of 5 EV Figures can be typeset. EV Figures should be cited as 'Figure EV1, Figure EV2' etc... in the text and their respective legends should be included in the main text after the legends of regular figures.

7) We would also encourage you to include the source data for figure panels that show essential data.

Numerical data should be provided as individual .xls or .csv files (including a tab describing the data). For blots or microscopy, uncropped images should be submitted (using a zip archive if multiple images need to be supplied for one panel). Additional information on source data and instruction on how to label the files are available .

8) Regarding data quantification, please ensure to specify the name of the statistical test used to generate error bars and P values, the number (n) of independent experiments underlying each data point (not replicate measures of one sample), and the test used to calculate p-values in each figure

legend. Discussion of statistical methodology can be reported in the materials and methods section, but figure legends should contain a basic description of n, P and the test applied.

Please note that error bars and statistical comparisons may only be applied to data obtained from at least three independent biological replicates.

I look forward to seeing a revised version of your manuscript when it is ready. Please let me know if you have questions or comments regarding the revision.

Kind regards,

Deniz

Deniz Senyilmaz Tiebe, PhD
Editor
EMBO Reports

Referee #1:

The paper describes the role of Cdk4 in the regulation of brown adipocyte function. The authors demonstrate that global loss of Cdk4 leads to increased browning, cold resistance as well as enhanced energy expenditure. The authors nicely show that the effect is due to beta-adrenergic signaling but seems to be non-cell autonomous since specific ablation of Cdk4 in BAT does not phenocopy the global knockout mouse. In a VMH specific Cdk4 ko mouse the higher fiber density observed in the global ko mice is phenocopied suggesting that Cdk4 controls BAT activity by regulation of sympathetic innervation.

There are several points that need to be addressed:

1. The introduction is quite long, the authors should shorten it, especially the summary at the end which is in essence the abstract
2. The regulation of beta-3-receptor mRNA in the inguinal WAT should be treated with caution. Data indicates that prolonged activation leads to a reduction rather than an increase in the receptor mRNA. If the authors want to base some arguments on this regulation the protein levels and localizations would need to be shown.
3. Fig.4 and Fig. 5 are TH protein levels increased in ingWAT and scWAT?
4. The data from Fig. 5 is a bit counter-intuitive. The authors observe the same increase in TH fibers but no change in thermogenesis. This obviously makes it difficult to reconcile the observations. Did they check in ingWAT, possibly under stimulated conditions if thermogenesis is altered. Are the compensatory reductions, which could explain the lack of a response (beta-receptors for example). The argument of embryonic deletion does not really fit as the global knockout would probably recombine even earlier than the VMH specific one.

Referee #2:

In the manuscript "Hypothalamic CDK4 regulates thermogenesis by modulating the sympathetic innervation and activation of brown adipose tissue," Castillo-Armengol et al. propose a role for CDK4 in the regulation of thermogenesis via SF1 neurons of the ventromedial hypothalamus. They demonstrate that CDK4 mice have higher energy expenditure under standard housing conditions and respond better to cold challenge. This is explained by increased activity of the brown adipose tissue. While knockout of CDK4 in BAT does not phenocopy whole body knockout, knockout of CDK4 in SF1 neurons in the VMH is able to partially reproduce the phenotype, suggesting a role of CDK4 in hypothalamic control of metabolism.

This work demonstrates a previously unknown role for CDK4, more thoroughly studied as a cell-cycle regulatory protein, in control of metabolism. Although there are publications exploring the role of CDK4 in muscle and adipose tissue, this is the first to demonstrate role for this protein in CNS control of metabolism. These observations will be of interest to those following the increasing number of publications regarding this topic. However, there are still some issues that must be addressed.

1. In the analysis of indirect calorimetry data from the CLAMS system, the authors use averages of VO₂, EE, and RER to compare groups. While the differences they observe are valid and statistically significant, a more nuanced and complete analysis could be done using ANCOVA. This would be easily done with publicly available software such as CalR.
2. The Western blots in the paper are normalized to Ponceau stain, which is less sensitive than a loading control such as Actin or GAPDH. Such a loading control should be included.
3. In figures 4 and 5, the authors demonstrate that CDK4 knockout results in an increase in TH-positive fibers in BAT. Although the IHC images are accompanied by a quantitation of these fibers, the validity of this data is still subject to the sections of tissue chosen for analysis. A bulk analysis of the BAT depot using Western blot should be performed to show an increase in TH overall. It would also be informative to demonstrate an increase in phosphorylated hormone sensitive lipase by Western, as would be expected with an increase in adrenergic signaling in this tissue.
4. Finally, the authors should more thoroughly address the issue that knockout of CDK4 in the SF1 neurons of the VMH does not fully recapitulate the whole-body knockout phenotype. While the SF1-Cre mice show resistance to cold exposure, they do not exhibit the differences in BAT morphology or gene expression or the changes in energy expenditure, food intake, and indirect calorimetry measurements that the whole-body CDK4 knockout mice exhibit. While the authors hypothesize in the discussion that this could be due to developmental effects, it seems more likely that SF1 neurons of the VMH are only partially responsible for the phenotype. Other regions of the CNS, for example, could also contribute, as could elements of the peripheral nervous system. This should at the very least be addressed in the discussion. Experiments with a pan-neuronal Cre, such as Syn1 Cre, would effectively address the question of whether neuronal CDK4 is entirely responsible for the phenotypes observed in the global CDK4 knockout. Alternatively, injection of AAV-Cre into specific brain regions could be used to identify other nuclei involved in regulation of thermogenesis by CDK4.

Referee #1:

The paper describes the role of Cdk4 in the regulation of brown adipocyte function. The authors demonstrate that global loss of Cdk4 leads to increased browning, cold resistance as well as enhanced energy expenditure. The authors nicely show that the effect is due to beta-adrenergic signaling but seems to be non-cell autonomous since specific ablation of Cdk4 in BAT does not phenocopy the global knockout mouse. In a VMH specific Cdk4 ko mouse the higher fiber density observed in the global ko mice is phenocopied suggesting that Cdk4 controls BAT activity by regulation of sympathetic innervation.

We thank this reviewer for the comments and suggestions. We have performed several new experiments to prove the physiological relevance of our findings. We have addressed the major concerns and suggestions as follows:

There are several points that need to be addressed:

1. The introduction is quite long, the authors should shorten it, especially the summary at the end which is in essence the abstract.

Following the reviewer's suggestion, we have now reviewed the introduction and done the appropriate changes to shorten and improve it.

2. The regulation of beta-3-receptor mRNA in the inguinal WAT should be treated with caution. Data indicates that prolonged activation leads to a reduction rather than an increase in the receptor mRNA. If the authors want to base some arguments on this regulation the protein levels and localizations would need to be shown.

To address this question, and following this reviewer suggestion, we have done two experiments. First, whole tissue lysates of scWAT were analyzed by Western blotting to assess the beta-3-adrenergic receptor (ADRB3) levels of expression. This experiment indicated no significant differences between scWAT of *Cdk4*^{+/+} and *Cdk4*^{-/-} mice (Appendix Fig S1A-B). On the other hand, we have performed a tissue subcellular fractionation of the same samples to determine the location of ADRB3 protein. This experiment clearly showed that ADRB3 was located at the plasma membrane in the scWAT of both *Cdk4*^{+/+} and *Cdk4*^{-/-} mice (Appendix Fig S1C). In conclusion, despite our previous experiment showed that *Adrb3* gene expression was significantly increased in the scWAT of *Cdk4*^{-/-} mice, no differences were found at the level of protein expression. The lack of differences, together with our results showing no differences in TH protein expression in the tissue (see next question), suggest that the adrenergic pathway is not changed in the white adipose tissue of the *Cdk4*^{-/-} mice.

Appendix Figure S1

3. Fig.4 and Fig. 5 are TH protein levels increased in ingWAT and scWAT?

We have performed immunohistochemical analysis of TH protein in the scWAT of both models of *Cdk4* deletion (*Cdk4*^{-/-} and *Cdk4*^{flx/flx} *Sf1-Cre* mice). We have observed no differences in TH protein levels in the scWAT of *Cdk4*^{-/-} mice (Appendix Fig S2A-B). In contrast, a higher amount of TH positive fibers was found in the *Cdk4*^{flx/flx} *Sf1-Cre* compared to the control *Cdk4*^{flx/flx} scWAT (Fig EV5B-C). This result was confirmed by the protein quantification resulting from a western blot analysis of TH (Fig EV5D-E). These new findings suggest that in the *Cdk4*^{flx/flx} *Sf1-Cre* there is an overall increase in the sympathetic innervation in adipose tissue and that both BAT and scWAT are contributing to the enhanced cold resistance of these animals.

Appendix Figure S2

Figure EV5

4. The data from Fig. 5 is a bit counter-intuitive. The authors observe the same increase in TH fibers but no change in thermogenesis. This obviously makes it difficult to reconcile the observations. Did they check in ingWAT, possibly under stimulated conditions if thermogenesis is altered. Are the compensatory reductions, which could explain the lack of a response (beta-receptors for example). The argument of embryonic deletion does not really fit as the global knockout would probably recombine even earlier than the VMH specific one.

We agree with this reviewer that increased innervation of the tissue (TH) should result in changes in gene expression in BAT. We could not find, however differences in the expression of thermogenesis genes in this tissue. But consistent with increased TH, these mice were more resistant to cold. As this reviewer proposes, changes in WAT could, at least partially, explain this paradox. Following reviewer's suggestion, we have analyzed the expression levels of thermogenic genes in the scWAT of the *Cdk4^{flox/flox} Sf1-Cre* mice in basal conditions. We found a tendency to increase thermogenic gene markers such as *Ucp1* or *Dio2*, and a significant increase in the expression of the thermogenic gene marker *Prdm16*, essential for brown and beige adipocyte identity and function (Fig EV5A). Most important is the finding that the protein expression of UCP1 was highly increased in the scWAT of the *Cdk4^{flox/flox} Sf1-Cre* mice indicating and confirming that these animals present a recruitment of thermogenic adipocytes in the scWAT depot in order to defeat thermal stress (Fig EV5D, F).

Figure EV5

Referee #2:

In the manuscript "Hypothalamic CDK4 regulates thermogenesis by modulating the sympathetic innervation and activation of brown adipose tissue," Castillo-Armengol et al. propose a role for CDK4 in the regulation of thermogenesis via SF1 neurons of the ventromedial hypothalamus. They demonstrate that CDK4 mice have higher energy expenditure under standard housing conditions and respond better to cold challenge. This is explained by increased activity of the brown adipose tissue. While knockout of CDK4 in BAT does not phenocopy whole body knockout, knockout of CDK4 in SF1 neurons in the VMH is able to partially reproduce the phenotype, suggesting a role of CDK4 in hypothalamic control of metabolism.

This work demonstrates a previously unknown role for CDK4, more thoroughly studied as a cell-cycle regulatory protein, in control of metabolism. Although there are publications exploring the role of CDK4 in muscle and adipose tissue, this is the first to demonstrate role for this protein in CNS control of metabolism. These observations will be of interest to those following the increasing number of publications regarding this topic. However, there are still some issues that must be addressed.

We would like to thank the reviewer for the positive evaluation of our manuscript.

1. In the analysis of indirect calorimetry data from the CLAMS system, the authors use averages of VO₂, EE, and RER to compare groups. While the differences they observe are valid and statistically significant, a more nuanced and complete analysis could be done using ANCOVA. This would be easily done with publicly available software such as CalR.

As suggested by the reviewer, we have now used the Cal R software (Mina, LeClair et al., 2018) to use the generalized linear model (GLM), utilizing body mass as a covariate to model mass-dependent metabolic parameters obtained with indirect calorimetry. The results of these novel analysis are now found in Figure 1G-I, and figure EV4I-J. Both the "genotype effect" (or "group effect") and the "mass effect" are now shown. A clear significant genotype difference was identified for energy balance, with *Cdk4*^{-/-} mice clearly showing a negative energy balance at room temperature, in agreement with their increased oxidative capacity, thermogenesis and sympathetic tone, and a near significant energy balance difference between *Cdk4*^{+/+} mice and *Cdk4*^{-/-} mice at 6°C; and this despite a modest sample size (N=6-5).

Figure 1

I

Effect	24°C		6°C		GLM
	Mass	Group	Mass	Group	
Oxygen Consumption (ml/hr)	0.8669	0.9905	0.5518	0.7005	
Carbon Dioxide Production (ml/hr)	0.7412	0.9294	0.5739	0.8	
Energy Expenditure (kcal/hour)	0.8386	0.9772	0.5564	0.7218	
Energy Balance (kcal/hour)	0.0222*	0.0283*	0.0307*	0.0656	

Effect	24°C		6°C		ANOVA
	Group	Group	Group	Group	
Respiratory Exchange Ratio		0.2058		<0.001**	

Figure EV4

RER, a mass independent metabolic parameter was analyzed using ANOVA. A *p-value* of 0.2058 was obtained at room temperature and a *p-value* of <0.001 *** was obtained at 6°C.

However, no significant differences were identified using GLM when total body mass was used a covariate for oxygen consumption and energy expenditure.

We have chosen to depict these mass-dependent metabolic variables normalized to body weight, due to dramatic differences of body mass observed between *Cdk4*^{+/+} mice and *Cdk4*^{-/-} mice (*Cdk4*^{-/-} mice weigh on average 9 grams less than *Cdk4*^{+/+} mice, meaning roughly 30% less).

2. The Western blots in the paper are normalized to Ponceau stain, which is less sensitive than a loading control such as Actin or GAPDH. Such a loading controls should be included.

We have performed additional western blot analysis to add the appropriate loading controls for each experiment. HSP90, GAPDH or TUBULIN have been used for this purpose. The newly included loading controls can be found in Fig 2D, Fig 3F, Fig 6H, Fig EV3C, Fig EV5D.

3. In figures 4 and 5, the authors demonstrate that CDK4 knockout results in an increase in TH-positive fibers in BAT. Although the IHC images are accompanied by a quantitation of these fibers, the validity of this data is still subject to the sections of tissue chosen for analysis. A bulk analysis of the BAT depot using Western blot should be performed to show an increase in TH overall. It would also be informative to demonstrate an increase in phosphorylated hormone sensitive lipase by Western, as would be expected with an increase in adrenergic signaling in this tissue.

We thank this reviewer for this suggestion. We have now analyzed the protein expression of TH in BAT protein lysates of equivalent protein concentration and found, surprisingly, a decrease between *Cdk4*^{-/-} and *Cdk4*^{+/+} mice (Appendix Fig S3A-B).

Appendix Fig S3

This result is in conflict with the previous reported observations of increase TH positive fibers by immunohistochemistry performed in the same tissues. It is important to mention that our immunohistochemistry studies are 10 representative photos of 2-3 sections for each mouse, covering a representative surface of the whole tissue. Also, the IHC quantification of TH positive fibers is normalized to 100 adipocytes, meaning that the increase in the sympathetic fiber content of the *Cdk4*^{-/-} mice is a measure of the stimulation that the thermogenic cells can receive (Fig 4A-B). We then asked for the underlying reason that could explain the discrepancy between western blot (WB) and immunohistochemistry (IHC) results. Indeed, the morphology and structure of the BAT of *Cdk4*^{+/+} and *Cdk4*^{-/-} mice is very different. As seen on images of Figure 4A, lipid content is strikingly decreased in *Cdk4*^{-/-} mice, leading to a higher amount of proteins per amount of tissue (Appendix Fig S3E). WB analyses are normalized by tissue protein content, whereas IHC results are normalized by number of adipocytes. Hence, considering one similar

nerve fiber innervating one adipocyte, the ratio of TH content to the protein will be artificially increased in *Cdk4*^{+/+} mice and decreased in *Cdk4*^{-/-} mice. In order to overcome this issue, we have analyzed the expression of TH protein normalized by the tissue weight (i.e., we loaded each well with the amount of proteins corresponding to the same amount of tissue). In these conditions, we found a clear increase in TH expression in the BAT of *Cdk4*^{-/-} mice when compared to an equal amount of BAT from the *Cdk4*^{+/+} mice (Appendix Fig S3F-G). Similar results were found for the synaptic protein PSD95 (Appendix Fig S3F, H), as well as for UCP1 (Fig S3F, I).

Appendix Fig S3

It is important to take into account that the IHC quantification of TH positive fibers is normalized to 100 adipocytes, meaning that the increase in the sympathetic fiber content of the *Cdk4*^{-/-} mice is a measure of the stimulation that the thermogenic cells can receive (Fig 4A-B). According to this experiment, the brown adipocytes in the BAT of *Cdk4*^{-/-} mice are more prone to be stimulated because they have more sympathetic fibers in their proximity compared to the adipocytes of the BAT from the *Cdk4*^{+/+} mice. In summary, our data shows that the BAT of *Cdk4*^{-/-}, considered as a whole tissue, is more innervated and has an increased TH content than the BAT of *Cdk4*^{+/+} mice.

As requested by the reviewer, we have also analyzed the expression levels of the phosphorylated hormone-sensitive lipase protein (p-HSL^{S660}) in BAT protein lysates of equivalent protein concentration. We observed no significant differences in the levels of expression of p-HSL^{S660} (Appendix Fig S3C-D). This controversial result can be explained by the fact that as the BAT of the *Cdk4*^{-/-} is delipidated in basal conditions, there might be a compensatory inhibition of HSL activity.

Appendix Figure S3

We have also analyzed the same parameters in the *Cdk4*^{flx/flx} *Sf1-Cre* mice. First, we have explored the TH protein expression by western blot in the BAT of these animals in basal conditions. The expression of TH protein was increased in the *Cdk4*^{flx/flx} *Sf1-Cre* mice, in accordance with the immunohistochemical analysis performed before (Appendix Fig S4A-B). Interestingly, the morphology and structure of the BAT of these mice is identical to the BAT of control mice.

Appendix Figure S4

Nevertheless, the analysis of phosphorylated hormone sensitive lipase (p-HSL^{S660}) in the same animals showed a big variability in the samples and no differences in the expression of this protein (Appendix Fig S4C-D). This controversial result could be explained by the fact that the animals had a minimum thermal stress in these conditions (RT).

Appendix Figure S4

4. Finally, the authors should more thoroughly address the issue that knockout of CDK4 in the SF1 neurons of the VMH does not fully recapitulate the whole-body knockout phenotype. While the SF1-Cre mice show resistance to cold exposure, they do not exhibit the differences in BAT morphology or gene expression or the changes in energy expenditure, food intake, and indirect

calorimetry measurements that the whole-body CDK4 knockout mice exhibit. While the authors hypothesize in the discussion that this could be due to developmental effects, it seems more likely that SF1 neurons of the VMH are only partially responsible for the phenotype. Other regions of the CNS, for example, could also contribute, as could elements of the peripheral nervous system. This should at the very least be addressed in the discussion. Experiments with a pan-neuronal Cre, such as Syn1 Cre, would effectively address the question of whether neuronal CDK4 is entirely responsible for the phenotypes observed in the global CDK4 knockout. Alternatively, injection of AAV-Cre into specific brain regions could be used to identify other nuclei involved in regulation of thermogenesis by CDK4.

We particularly thank this reviewer for this suggestion. We agree that it is important to determine if *Cdk4* deletion modulates neuronal activity in other thermogenic brain nuclei which could explain why the phenotype of the *Cdk4^{flox/flox} Sf1-Cre* mice does not exactly replicate the phenotype of the *Cdk4^{-/-}* mice. To address this hypothesis, we have evaluated the neuronal activity in the brain of *Cdk4^{-/-}* mice by assessing the expression of c-FOS, which is a transcription factor whose expression is enhanced upon neuronal depolarization and it has been broadly used as a marker of neuronal activation (Bullitt, 1990, West, Griffith et al., 2002). Therefore, we quantified the number of c-FOS immunoreactive cells in the brain of *Cdk4^{-/-}* and *Cdk4^{+/+}* mice in basal conditions paying special attention to the medial preoptic area (mPOA) (Morrison, 2016, Morrison, Madden et al., 2014, Yoshida, Nakamura et al., 2003), the arcuate (ARC) (Rahmouni & Morgan, 2007), the ventromedial hypothalamus (VMH) (Kim, Zhao et al., 2011, Seoane-Collazo, Roa et al., 2018), the dorsomedial hypothalamus (DMH) (Hogan, Coscina et al., 1982, Monge-Roffarello, Labbe et al., 2014) and the lateral hypothalamic area (LH) (Martins, Seoane-Collazo et al., 2016, Sellayah, Bharaj et al., 2011); all well-known areas involved in the response to thermal stress. Immunofluorescence analysis showed that *Cdk4^{-/-}* mice have a significant increment of c-FOS positive cells in the VMH when compared to the *Cdk4^{+/+}* control mice. This data is consistent with our previous findings in the *Cdk4^{flox/flox} Sf1-Cre* mice which indicated that CDK4 is an important regulator of the VMH activity.

Interestingly, a significant higher staining of c-FOS positive cells was also found in the ARC and LHA and it is important to mention that a high tendency to an increased c-FOS expression was also found in the mPOA of the *Cdk4^{-/-}* mice. This is a very exciting finding, not only because it indicates that the phenotype of the *Cdk4^{-/-}* mice could be explained by the relevant participation of CDK4 in other brain regions that might be also regulating thermogenesis, but also because it points to a new role of CDK4 in general homeostasis in the brain.

The representative images and quantification of this experiment have been added to the manuscript in new Figure 5. These new findings were also added in the discussion of the revised version of this manuscript.

New figure 6

References

Bullitt E (1990) Expression of c-fos-like protein as a marker for neuronal activity following noxious stimulation in the rat. *J Comp Neurol* 296: 517-30

Hogan S, Coscina DV, Himms-Hagen J (1982) Brown adipose tissue of rats with obesity-inducing ventromedial hypothalamic lesions. *Am J Physiol* 243: E338-44

Kim KW, Zhao L, Donato J, Jr., Kohno D, Xu Y, Elias CF, Lee C, Parker KL, Elmquist JK (2011) Steroidogenic factor 1 directs programs regulating diet-induced thermogenesis and leptin action in the ventral medial hypothalamic nucleus. *Proc Natl Acad Sci U S A* 108: 10673-8

Martins L, Seoane-Collazo P, Contreras C, Gonzalez-Garcia I, Martinez-Sanchez N, Gonzalez F, Zalvide J, Gallego R, Dieguez C, Nogueiras R, Tena-Sempere M, Lopez M (2016) A Functional Link between AMPK and Orexin Mediates the Effect of BMP8B on Energy Balance. *Cell Rep* 16: 2231-2242

Mina AI, LeClair RA, LeClair KB, Cohen DE, Lantier L, Banks AS (2018) CalR: A Web-Based Analysis Tool for Indirect Calorimetry Experiments. *Cell Metab* 28: 656-666 e1

Monge-Roffarello B, Labbe SM, Lenglos C, Caron A, Lanfray D, Samson P, Richard D (2014) The medial preoptic nucleus as a site of the thermogenic and metabolic actions of melanotan II in male rats. *Am J Physiol Regul Integr Comp Physiol* 307: R158-66

Morrison SF (2016) Central control of body temperature. *F1000Res* 5

Morrison SF, Madden CJ, Tupone D (2014) Central neural regulation of brown adipose tissue thermogenesis and energy expenditure. *Cell Metab* 19: 741-756

Rahmouni K, Morgan DA (2007) Hypothalamic arcuate nucleus mediates the sympathetic and arterial pressure responses to leptin. *Hypertension* 49: 647-52

Sellayah D, Bharaj P, Sikder D (2011) Orexin is required for brown adipose tissue development, differentiation, and function. *Cell Metab* 14: 478-90

Seoane-Collazo P, Roa J, Rial-Pensado E, Linares-Pose L, Beiroa D, Ruiz-Pino F, Lopez-Gonzalez T, Morgan DA, Pardavila JA, Sanchez-Tapia MJ, Martinez-Sanchez N, Contreras C, Fidalgo M, Dieguez C, Coppari R, Rahmouni K, Nogueiras R, Tena-Sempere M, Lopez M (2018) SF1-Specific AMPK α 1 Deletion Protects Against Diet-Induced Obesity. *Diabetes* 67: 2213-2226

West AE, Griffith EC, Greenberg ME (2002) Regulation of transcription factors by neuronal activity. *Nat Rev Neurosci* 3: 921-31

Yoshida K, Nakamura K, Matsumura K, Kanosue K, Konig M, Thiel HJ, Boldogkoi Z, Toth I, Roth J, Gerstberger R, Hubschle T (2003) Neurons of the rat preoptic area and the raphe pallidus nucleus innervating the brown adipose tissue express the prostaglandin E receptor subtype EP3. *Eur J Neurosci* 18: 1848-60

Dear Lluís,

Thank you for submitting the revised version of your manuscript. It has now been seen by both of the original referees.

As you can see, the referees find that the study is significantly improved during revision and recommend publication. Before I can accept the manuscript, I need you to address some minor points below:

- Please comment on the overexposure issue of the tubulin blot of EV Figure 2A in the figure legend. Please upload the image exported from the Western Blot developer that shows the saturation points as source data and refer to it in the figure legend, as well.
- As per our guidelines, please add a 'Data Availability Section', where you state that no data were deposited in a public database.
- We noted that Wiebe Venema's initials are misspelled as VB in Author Contributions.
- We realized that the funding information is incomplete in the manuscript submission system.
- We noted that the appendix figures are currently not called out in the text.
- Papers published in EMBO Reports include a 'Synopsis' to further enhance discoverability. Synopses are displayed on the html version of the paper and are freely accessible to all readers. The synopsis includes a short standfirst summarizing the study in 1 or 2 sentences that summarize the key findings of the paper and are provided by the authors and streamlined by the handling editor. I would therefore ask you to include your synopsis blurb.
- In addition, please provide an image for the synopsis. This image should provide a rapid overview of the question addressed in the study but still needs to be kept fairly modest since the image size cannot exceed 550x400 pixels.
- Our production/data editors have asked you to clarify several points in the figure legends (see attached document). Please incorporate these changes in the attached word document and return it with track changes activated.

Thank you again for giving us to consider your manuscript for EMBO Reports, I look forward to your minor revision.

Kind regards,

Deniz

--

Deniz Senyilmaz Tiebe, PhD
Editor
EMBO Reports

Referee #1:

The authors addressed all my concerns very convincingly and I would suggest to accept the paper

Referee #2:

The revised paper is acceptable for publication.

Dear Deniz,

First of all, I want to thank you for the overall process of review of our manuscript. We appreciate the comments of both the reviewers and the editor. We are now sending a revised version addressing the minor points raised in the last review.

- Please comment on the overexposure issue of the tubulin blot of EV Figure 2A in the figure legend. Please upload the image exported from the Western Blot developer that shows the saturation points as source data and refer to it in the figure legend, as well.

We explain now in the figure legend the issue of saturation. We refer to the file from the imager that we uploaded in the data source folder.

- As per our guidelines, please add a 'Data Availability Section', where you state that no data were deposited in a public database.

We added the Data availability section

- We noted that Wiebe Venema's initials are misspelled as VB in Author Contributions.

This has been corrected

- We realized that the funding information is incomplete in the manuscript submission system.

We included all funding information in the online submission system

- We noted that the appendix figures are currently not called out in the text.

We are now referring to the appendix figures in the text.

- Papers published in EMBO Reports include a 'Synopsis' to further enhance discoverability. Synopses are displayed on the html version of the paper and are freely accessible to all readers. The synopsis includes a short standfirst summarizing the study in 1 or 2 sentences that summarize the key findings of the paper and are provided by the authors and streamlined by the handling editor. I would therefore ask you to include your synopsis blurb.

We have uploaded the synopsis blurb

- In addition, please provide an image for the synopsis. This image should provide a rapid overview of the question addressed in the study but still needs to be kept fairly modest since the image size cannot exceed 550x400 pixels.

We have uploaded a synopsis image

- Our production/data editors have asked you to clarify several points in the figure legends (see attached document). Please incorporate these changes in the attached word document and return it with track changes activated.

We clarified and addressed the points raised by the production editors.

Dear Lluís,

Thank you for submitting your revised manuscript. I have now looked at everything and all looks fine. Therefore I am very pleased to accept your manuscript for publication in EMBO Reports.

Congratulations on a nice work!

Kind regards,

Deniz

--

Deniz Senyilmaz Tiebe, PhD
Editor
EMBO Reports

THINGS TO DO NOW:

You will receive proofs by e-mail approximately 2-3 weeks after all relevant files have been sent to our Production Office; you should return your corrections within 2 days of receiving the proofs.

Please inform us if there is likely to be any difficulty in reaching you at the above address at that time. Failure to meet our deadlines may result in a delay of publication, or publication without your corrections.

All further communications concerning your paper should quote reference number EMBOR-2019-49807V3 and be addressed to emboreports@wiley.com.

Should you be planning a Press Release on your article, please get in contact with emboreports@wiley.com as early as possible, in order to coordinate publication and release dates.